# Hybrid Quantum-Classical Recurrent Neural Networks

## Abstract

We present a hybrid quantum-classical recurrent neural network (QRNN) architecture in which the recurrent core is realized as a parametrized quantum circuit (PQC) controlled by a classical feedforward network. The hidden state is the quantum state of an $n$-qubit PQC in an exponentially large Hilbert space $\mathbb{C}^{2^n}$, which serves as a coherent recurrent quantum memory. The PQC is unitary by construction, making the hidden-state evolution norm-preserving without external constraints. At each timestep, mid-circuit Pauli expectation-value readouts are combined with the input embedding and processed by the feedforward network, which provides explicit classical nonlinearity. The outputs parametrize the PQC, which updates the hidden state via unitary dynamics. The QRNN is compact and physically consistent, and it unifies (i) unitary recurrence as a high-capacity memory, (ii) partial observation via mid-circuit readouts, and (iii) nonlinear classical control for input-conditioned parametrization. We evaluate the model in simulation with up to 14 qubits on sentiment analysis, MNIST, permuted MNIST, copying memory, and language modeling. For sequence-to-sequence learning, we further devise a soft attention mechanism over the mid-circuit readouts and show its effectiveness for machine translation. To our knowledge, this is the first model (RNN or otherwise) grounded in quantum operations to achieve competitive performance against strong classical baselines across a broad class of sequence-learning tasks.

## 1 Introduction

Recurrent neural networks (RNNs) process sequence data by maintaining a hidden state that is updated at each timestep, which can create a bottleneck for memory and representational capacity. While vanilla RNNs have been empirically shown to retain roughly one real value of information per hidden unit, with the effective task-specific capacity linearly bounded by the number of model parameters (Collins et al., 2017), similar limitations extend to gated architectures such as LSTMs and GRUs (Hochreiter and Schmidhuber, 1997; Cho et al., 2014), despite their use of gating and explicit memory cells (Collins et al., 2017). This means that more complex sequences may exceed what the hidden state can encode, forcing the model to compress or forget.

Another challenge in training RNNs is the vanishing and exploding gradient problem (Bengio et al., 1994; Hochreiter and Schmidhuber, 1997), which arises from repeated multiplication through the recurrent Jacobian. Among various strategies to address this (Mikolov, 2012; Pascanu et al., 2013; Le et al., 2015), unitary and orthogonal RNNs (Arjovsky et al., 2016; Jing et al., 2019; Helfrich et al., 2018; Kiani et al., 2022) constrain the recurrent weights to be norm-preserving, allowing gradients to remain stable across timesteps.

The introduction of the Transformer (Vaswani et al., 2017) appeared to obviate explicit recurrence by bypassing the hidden-state bottleneck. However, recent work shows that recurrent inductive bias remains highly competitive and provides representational advantages not matched by Transformers (Gu and Dao, 2023; Orvieto et al., 2023; Bhattamishra et al., 2024; Beck et al., 2024).

With the advancement of quantum computing (Arute et al., 2019; Acharya et al., 2024; Reichardt et al., 2024; DeCross et al., 2025), parametrized quantum circuits (PQCs), which are a core component of variational hybrid quantum-classical models, have concurrently emerged as an alternative mechanism for function approximation (Benedetti et al., 2019; Du et al., 2019; Bondesan and Welling, 2020; Pérez-Salinas et al., 2021; Schuld et al., 2021; Yu et al., 2024b). PQCs implement unitary transformations

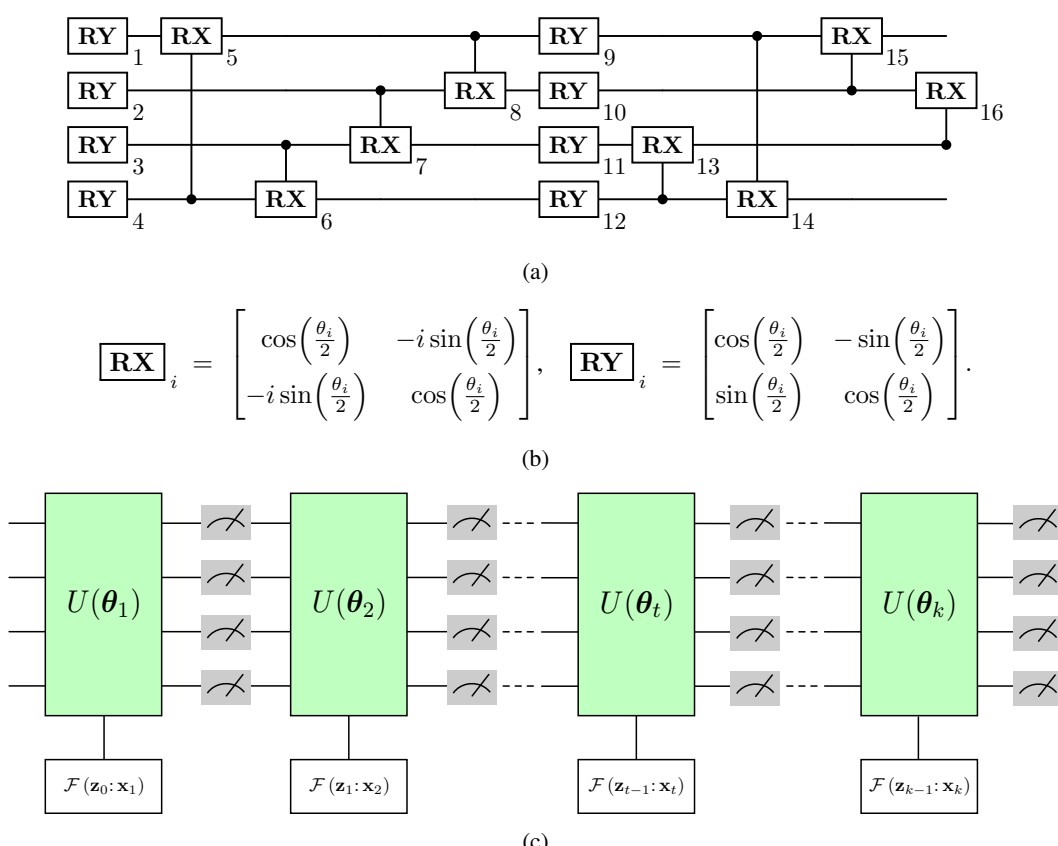

$$\boxed{\text{RX}}_i = \begin{bmatrix} \cos\left(\frac{\theta_i}{2}\right) & -i\sin\left(\frac{\theta_i}{2}\right) \\ -i\sin\left(\frac{\theta_i}{2}\right) & \cos\left(\frac{\theta_i}{2}\right) \end{bmatrix}, \quad \boxed{\text{RY}}_i = \begin{bmatrix} \cos\left(\frac{\theta_i}{2}\right) & -\sin\left(\frac{\theta_i}{2}\right) \\ \sin\left(\frac{\theta_i}{2}\right) & \cos\left(\frac{\theta_i}{2}\right) \end{bmatrix}.$$

(b)

(c)

Figure 1: Hybrid QRNN. (a) Recurrent core PQC with $n = 4$ qubits (illustrative) and 16 parametrized gates, acting on a quantum state in the Hilbert space $\mathbb{C}^{2^n}$; each horizontal line corresponds to one qubit. (b) **RX** and **RY** gates.[1] Each gate in the PQC is parametrized by a rotation angle $\theta_i$, where $1 \leq i \leq 16$. (c) QRNN unrolled for a sequence of length $k$. The feedforward network $\mathcal{F}$ takes as input the concatenation of the mid-circuit readout vector from the previous timestep and the current input $(\mathbf{z}_{t-1} : \mathbf{x}_t)$, and outputs $\boldsymbol{\theta}_t \in \mathbb{R}^{16}$ containing the 16 rotation angles that parametrize the PQC shown in (a) as $U(\boldsymbol{\theta}_t)$ (usually referred to as *angle encoding*),[2] which acts on the quantum state propagated from the previous step. ⟋ denotes qubit expectation-value readouts.

by construction, which naturally preserve norms (§3.1). Acting on $n$ qubits, they enable expressive transformations over quantum states in an exponentially large Hilbert space $\mathbb{C}^{2^n}$. Although such spaces are classically intractable beyond moderate $n$, they can be manipulated with only $n$ qubits on quantum hardware.

In this work, we present a hybrid quantum-classical QRNN in which the recurrent core is realized as a PQC controlled by a classical feedforward network. The hidden state is the quantum state of the $n$-qubit PQC, residing in an exponentially large Hilbert space $\mathbb{C}^{2^n}$, which provides a high-capacity coherent quantum memory. Nonlinearity is introduced through classical computation rather than approximated within the quantum circuit, leaving the PQC strictly for unitary evolution of the hidden state. Fig. 1 illustrates both the PQC (with four qubits shown for illustration) and the unrolled QRNN:

- At timestep $t$, the input is mapped to a learnable representation $\mathbf{x}_t$ via an embedding layer.
- A classical feedforward network $\mathcal{F}$ takes as input the concatenation of $\mathbf{z}_{t-1}$ (readout outputs at timestep $t-1$) and the current input $\mathbf{x}_t$. It outputs the PQC parameters $\boldsymbol{\theta}_t$ which configure the PQC with a fixed gate layout (Fig. 1a), denoted $U(\boldsymbol{\theta}_t)$, applied at timestep $t$ (Fig. 1c).

---

[1]All **RX** gates are controlled rotations that apply only when the connected control qubit is in the $|1\rangle$ state: $\mathbf{CRX}(\theta_i) = |0\rangle\langle 0| \otimes I + |1\rangle\langle 1| \otimes \mathbf{RX}(\theta_i)$.

[2]$U(\boldsymbol{\theta}_t)$ is a single unitary for encoding the outputs of $\mathcal{F}$ and transforming the state.

- The PQC applies the parametrized unitary gates to evolve the quantum state, yielding the updated state. Residing in an exponentially large Hilbert space, this state persists across timesteps and provides the model's core recurrent memory.

- The mid-circuit readout $\mathbf{z}_t$ (or the final readout at the end of the sequence) is a real-valued vector obtained from the quantum state via Pauli expectation-value readouts and is used: (i) as recurrent feedback $\mathbf{z}_{t-1}$ at timestep $t$, and (ii) as the input to task-specific classical layers.

We develop the models on GPUs, allowing us to simulate and train quantum recurrence via classical backpropagation, with the expectation that such models will become classically unsimulatable as the number of qubits increases. To our knowledge, this is the first model (RNN or otherwise) grounded in quantum operations demonstrated in classical simulation with up to 14 qubits across six realistic sequence-modeling tasks, achieving competitive performance with LSTM and the scaled Cayley orthogonal scoRNN designed for norm preservation (Helfrich et al., 2018). Experiments also show that classical nonlinear control and feedback are effective, with nonlinear variants outperforming their linear counterparts, and that the unitary quantum recurrent core maintains more stable gradients than LSTMs on sequences of up to 400 tokens (§4.6).

The QRNN is motivated in part by the memory and gradient problems of RNNs, but its main aim is to explore a hybrid quantum–classical recurrent model in an idealized proof of principle that allows us to study its computational behavior under best-case conditions across a broad class of sequence learning tasks. The PQC (Sim et al., 2019) uses only one- and two-qubit gates without nonstandard operations and is feasible on quantum hardware (Xu et al., 2024). The overall architecture provides a hardware-aware base case and a plausible path toward future hardware implementations, which would require fault-tolerant devices capable of sustaining long coherent recurrences and real-time classical control for realistic-scale sequence modeling.

Another way to view the QRNN is via fast and slow weights in RNNs, which function as different types of memory across multiple timescales (Schmidhuber, 1992; Ba et al., 2016). The PQC parameters serve as the short-term memory, analogous to the hidden activities of classical RNNs, and are controlled and reconfigured at each timestep by a classical feedforward network whose slow weights encode the long-term memory. The quantum state, updated via unitary transformations, evolves on a faster timescale than the slow weights, persists across timesteps, and acts as a third, higher-capacity memory in the Hilbert space, retaining information that influences subsequent computation (Hinton and Plaut, 1987; Schmidhuber, 1993).

## 2 RELATED WORK

Bausch (2020) proposes a QRNN whose recurrence is implemented by iterating a quantum cell built from "quantum neurons". Nonlinearity is induced *inside* the circuit via measurement-and-postselection primitives (repeat-until-success), together with amplitude amplification to make the procedure near-deterministic. Consequently, the per-timestep update is not strictly unitary and introduces back-action on the memory. Because the inherent linear dynamics of PQCs, the effective nonlinear maps achievable by such schemes are structurally constrained, and the repertoire of admissible nonlinearities remains limited (Yan et al., 2020; Moreira et al., 2023; Zi et al., 2024).

The so-called QLSTMs embed PQCs into the gating mechanisms of classical LSTMs (Chen et al., 2020; Yu et al., 2024a; Ubale et al., 2025), replacing dense layers in the LSTM gates with PQCs. However, all memory and recurrence remain entirely classical, governed by standard hidden and cell state updates. These architectures are best viewed as classical LSTMs augmented with auxiliary PQCs, rather than quantum recurrent models.

Li et al. (2023); Siemaszko et al. (2023) also model recurrences with PQCs and support per-timestep readouts, but they rely entirely on linear quantum dynamics without other nonlinearity or classical control. Nikoloska et al. (2023) stacked a PQC on top of a classical RNN, which modulates whether a quantum unitary is applied or skipped.[3]

Experiments with the existing models have focused on domain-specific tasks such as fraud detection (Ubale et al., 2025), low-resource text classification (Yu et al., 2024a), or scaled-down

---

[3]We experimented with a similar stacking architecture concurrently [link to code].

MNIST (Bausch, 2020; Siemaszko et al., 2023). We instead present the first QRNN to demonstrate competitive performance across six full-scale sequence modeling tasks.

## 3 MODEL

### 3.1 PQC

**Unitary evolution.** A PQC typically starts from the all-zero state $|\psi\rangle = |0\rangle^{\otimes n} \in \mathbb{C}^{2^n}$ and applies a series of gates arranged from left to right.[4] An example PQC with $n = 4$ qubits is shown in Fig. 1a, where each horizontal line represents a qubit. The square boxes denote quantum gates, which by definition are unitary transformations acting on one or more qubits. Single-qubit gates apply local transformations, while multi-qubit gates can generate superposition and entanglement.[5]

Let $U$ denote the composition (product) of a collection of unitary gates, hence $U^\dagger U = I$. For any state $|\psi\rangle$,

$$\|U|\psi\rangle\|^2 = \langle\psi|U^\dagger U|\psi\rangle = \langle\psi|I|\psi\rangle = \langle\psi|\psi\rangle = \|\psi\|^2,$$

which ensures norm preservation by construction.[6]

**Parametrized unitary gates.** In a PQC, gates can be either fixed or parametrized. Fixed gates implement structural operations and remain constant throughout training,[7] while the latter contain learnable parameters, which function like trainable weight matrices analogous to neural-network "layers". The PQC in Fig. 1a consists of entirely parametrized gates.

**Readouts via Pauli expectations.** A succinct way to summarize a quantum state is via *expectation values* of Pauli observables. For each qubit $k$, the Pauli operators $X_k, Y_k, Z_k$ are the three fundamental single-qubit observables that act on that qubit (with identity on all other qubits). For example, the Pauli-$Z$ operator measures the *computational basis* with

$$Z = \begin{pmatrix} 1 & 0 \\ 0 & -1 \end{pmatrix},$$

while $X$ and $Y$ measure superposition states in orthogonal bases. For a single qubit $|\psi\rangle = \alpha|0\rangle + \beta|1\rangle$, the Pauli-$Z$ expectation is $\langle Z\rangle = \langle\psi|Z|\psi\rangle = |\alpha|^2 - |\beta|^2 \in [-1, 1]$. For an $n$-qubit state, per-qubit expectations $\{\langle X_k\rangle, \langle Y_k\rangle, \langle Z_k\rangle\}_{k=1}^n$ and correlation terms (*Pauli strings*) $\langle P_1 \otimes \cdots \otimes P_n\rangle$ with $P_\ell \in \{I, X, Y, Z\}$ yield bounded, hardware-agnostic real-valued summaries useful for analysis and downstream modeling.

Although expectation-value readouts are nonlinear functions of parametrized rotation angles (e.g., in **RX** gates), the resulting nonlinearity is generally insufficient on its own (§4).

### 3.2 HYBRID MODEL

RNNs parametrize a conditional distribution with a function that depends on a hidden state $\mathbf{h}_{t-1}$, which compacts past inputs $(\mathbf{x}_1, \ldots, \mathbf{x}_{t-1})$ into a fixed-dimensional representation:

$$p(\mathbf{x}_t \mid \mathbf{x}_1, \ldots, \mathbf{x}_{t-1}) \approx p(\mathbf{x}_t \mid \mathbf{h}_{t-1}).$$

At each timestep $t$, the hidden state $\mathbf{h}_t$ is updated based on the previous hidden state $\mathbf{h}_{t-1}$ and the current input $\mathbf{x}_t$:

$$\mathbf{h}_t = f(\mathbf{h}_{t-1}, \mathbf{x}_t; \boldsymbol{\Theta}),$$

where $f$ is a transformation (e.g., a basic RNN or LSTM cell) parametrized by $\boldsymbol{\Theta}$. In the hybrid model (Fig. 1c), we replace the hidden state with a quantum state represented by the PQC in Fig. 1a, which is controlled by a classical feedforward network and evolved by applying the unitary gates.

---

[4]$\otimes$ denotes the tensor product.

[5]See Appendix D for a basic description of qubits and superposition.

[6]$U^\dagger$ denotes the conjugate transpose (Hermitian adjoint) of $U$. Formally, if the PQC consists of $L$ gates, $U = u_L u_{L-1} \cdots u_1$, where each $u_i$ is a unitary operator acting on some subset of qubits, then $U^\dagger = u_1^\dagger u_2^\dagger \cdots u_L^\dagger$, and hence $U^\dagger U = I$.

[7]For example, the **CNOT** gate flips the target qubit if the control is in the $|1\rangle$ state.

Let $\mathbf{x}_t$ be the input embedding at timestep $t$, and let $\mathbf{z}_{t-1}$ be the readout vector from the previous timestep. In the most generic form of the hybrid model,[8] the two are concatenated into a single vector $\mathbf{u}_t = (\mathbf{z}_{t-1} \colon \mathbf{x}_t)$ and passed through a classical feedforward network $\mathcal{F}$ with one hidden layer and a nonlinearity.

The first transformation in $\mathcal{F}$ maps the input $\mathbf{u}_t$ to a hidden representation $\mathbf{v}_t$:

$$\mathbf{v}_t = \phi(\mathbf{W}_1 \mathbf{u}_t + \mathbf{b}_1), \tag{1}$$

where $\phi$ is a nonlinear activation function. The second transformation maps $\mathbf{v}_t$ to

$$\boldsymbol{\theta}_t = \mathbf{W}_2 \mathbf{v}_t + \mathbf{b}_2, \tag{2}$$

where $\boldsymbol{\theta}_t \in \mathbb{R}^d$ represents the parameters that control the PQC's unitary operations at timestep $t$. Each element of $\boldsymbol{\theta}_t$ denoted $\theta_i$ is mapped to a rotation angle in a parametrized quantum gate within the PQC (e.g., $1 \leq i \leq d$ and $d = 16$ in Fig. 1a).

The PQC itself is defined by a unitary operator $U(\boldsymbol{\theta}_t)$ parametrized by $\boldsymbol{\theta}_t$.[9] Applying $U(\boldsymbol{\theta}_t)$ to the prior state $\mathbf{h}_{t-1} = |\psi_{t-1}\rangle$ yields the updated state

$$\mathbf{h}_t = U(\boldsymbol{\theta}_t) |\psi_{t-1}\rangle.$$

A classical readout vector is then computed from Pauli expectation values:

$$\mathbf{z}_t = \text{Readout}(\mathbf{h}_t) = \big( \langle X_1 \rangle_t, \; \langle Y_1 \rangle_t, \; \langle Z_1 \rangle_t, \; \ldots, \; \langle X_n \rangle_t, \; \langle Y_n \rangle_t, \; \langle Z_n \rangle_t \big), \tag{3}$$

where $\langle P_k \rangle_t = \langle \psi_t | P_k | \psi_t \rangle$ for $P_k \in \{X_k, Y_k, Z_k\}$ and $|\psi_t\rangle = \mathbf{h}_t$.[10]

Serving as a proxy for the quantum state, $\mathbf{z}_t$ is concatenated with the next input $\mathbf{x}_{t+1}$ to produce $\boldsymbol{\theta}_{t+1}$ via the classical controller. Because the Pauli expectation-value readouts do not alter the state, coherence of the quantum state is preserved across timesteps, allowing it to function as a coherent recurrent memory.

We train the entire hybrid model end-to-end using classical backpropagation, optimizing the parameters $\boldsymbol{\Theta} = \{\mathbf{W}_1, \mathbf{b}_1, \mathbf{W}_2, \mathbf{b}_2\}$ via standard optimizers, such as Adam (Kingma and Ba, 2014). For sequence-to-sequence learning, $\mathbf{z}_t$ provides per-timestep outputs and serves as contextual embeddings for soft attention decoding.

## 4    EXPERIMENTS

We use the ansatz shown in Fig.1a (scaled to more qubits when required) as the core circuit for the QRNN. Sim et al. (2019) demonstrate experimentally that this ansatz is expressive, capable of generating strong entanglement, and able to represent a significant portion of the Hilbert space, even compared to deeper circuits built from less expressive ansätze.[11] We implement and simulate the model using `TorchQuantum` (Wang et al., 2022), which remains less optimized than classical toolkits due to the lack of efficient kernels for hybrid operations involving tight classical–quantum feedback, particularly in recurrent settings. Our ansatz balances expressivity, implementation simplicity, and simulation efficiency.

For the feedforward network $\mathcal{F}$ (Eq. 1 and Eq. 2), we experimented with ReLU, LeakyReLU, GLU and GELU nonlinearities.[12] For both language modeling and translation, we first transform the readouts with a separate feedforward layer and use the result both for vocabulary classification and as input to the next timestep.

All experiments are run on a single A100/A30 GPU and we select the best models on the validation split across different random seeds and report the test results. The per-epoch training runtime ranges

---

[8]Extra transformations may be applied to the readouts before classifications or feeding them to the next step (for some tasks); see §4.

[9]Here $U(\boldsymbol{\theta}_t)$ denotes the product of the circuit's parametrized gates, each acting on one or more qubits with its parameter drawn from $\boldsymbol{\theta}_t$.

[10]For computational efficiency, we use only single-qubit expectations rather than multi-qubit Pauli strings.

[11]See Appendix E for details on the PQC design and expressibility evaluation methodology.

[12]GLU requires projecting to twice the output dimensionality, effectively increasing the parameter count compared to standard nonlinearities like ReLU, when all other dimensions are held constant.

Table 1: Classification accuracy on IMDB. Qubit count $q$, total readouts $m$; or hidden state size $h$ (for RNN, LSTM and scoRNN only); embedding dimension $e$; parameter count $p$. † indicates the LSTM in Dai and Le (2015).

| Model | Val | Test | $q_m \vee h$ | $e$ | $p$ |
|---|---|---|---|---|---|
| QRNN$_{\text{ReLU}}$ | 87.25 | 85.37 | $8_{24}$ | 100 | 5K |
| QRNN$_{\text{LeakyReLU}}$ | 87.41 | **87.00** | $8_{24}$ | 100 | 5K |
| QRNN$_{\text{GELU}}$ | 87.53 | 86.38 | $8_{24}$ | 100 | 5K |
| QRNN$_{\text{Linear}}$ | 85.37 | 84.21 | $8_{24}$ | 100 | 5K |
| QRNN$_{\text{Linear}}$ | 84.21 | 83.22 | $4_{12}$ | 100 | 2K |
| RNN | 87.64 | 86.96 | 50 | 50 | 5K |
| LSTM | 88.40 | 86.79 | 25 | 25 | 5.2K |
| LSTM$^{\dagger}$ | – | 86.5 | 1,024 | 512 | 6.2M |
| scoRNN | 87.25 | 86.48 | 50 | 50 | 5K |

from ~4 minutes for MNIST (with 10 qubits) to ~36 minutes for language modeling (with 14 qubits). Hyperparameters are tuned on the validation splits, the shared hyperparameters across all the tasks include the Adam optimizer without learning rate decay ($lr = 1 \times 10^{-3}$, $\lambda = 1 \times 10^{-4}$, and $\epsilon = 1 \times 10^{-10}$) and dropout applied to the input at each step, with task-dependent drop rates. We apply full-sequence backpropagation without truncation, except for language modeling, where sequences are truncated to 35 tokens. No pretrained word embeddings are used. Additional hyperparameters and test set statistics (mean, min, max across runs) are provided in Appendix C. For scoRNN, we use a hidden size of 170 and the hyperparameters from Helfrich et al. (2018) are used throughout.

## 4.1 SENTIMENT ANALYSIS

The IMDB sentiment dataset (Maas et al., 2011) is a balanced binary classification benchmark with 25K labeled reviews each for training and testing. The average review length is 241 tokens, with a maximum length of 2,500 tokens. We use 7.5K reviews from the training set for validation and truncate all reviews to a maximum length of 400 tokens across all models.

The hybrid model for this task follows the generic hybrid architecture described in §3.2. At the final input token, we apply an affine transformation to the readouts to produce two logits, which are used for classification via cross-entropy. Table 1 summarizes the results. QRNN$_{\text{LeakyReLU}}$ achieves the highest test accuracy. Ablating the classical nonlinearity (Eq. 1) degrades performance, though increasing the number of qubits in the linear model still yields some accuracy gains. Adding the nonlinearity results in a substantial improvement, outperforming all baselines. On this task, the orthogonal scoRNN underperforms other models, despite having a larger hidden state and over five times more parameters.

## 4.2 MNIST AND PERMUTED MNIST

We report results on the full MNIST dataset without downsampling using the same model as for IMDB, except with 10 output classes instead of binary classification. The standard pixel-by-pixel permuted MNIST (pMNIST) setup (Le et al., 2015; Arjovsky et al., 2016) requires 784 steps to process each $28 \times 28$ digit, which makes simulation prohibitively slow. Here we permute the pixels of each digit first, which are then reshaped back to $28 \times 28$. In both the standard and permuted cases, we use the same hyperparameters.

Table 2 shows that QRNNs with three different types of nonlinearity outperform the classical baselines on both tasks, clearly demonstrating the benefit of adding classical nonlinearities compared to the QRNN$_{\text{Linear}}$ models. We observe that permutation leads to an accuracy drop across all models: 2.45% for QRNN$_{\text{GELU}}$, 3.00% for the RNN, 3.51% for the LSTM, and 1.51% for scoRNN, which achieves comparable performance to QRNN$_{\text{GELU}}$.

Table 2: Classification accuracy on MNIST and pMNIST. Qubit count $q$, total readouts $m$; or hidden state size $h$ (for RNN, LSTM and scoRNN only); embedding dimension $e$; parameter count $p$. † indicates the QRNN model of (Bausch, 2020) with 13 qubits and each digit downscaled to $4 \times 4$ and binarized.

| | MNIST | | pMNIST | | | | |
|---|---|---|---|---|---|---|---|
| **Model** | **Val** | **Test** | **Val** | **Test** | $q_m \vee h$ | $e$ | $p$ |
| QRNN$_{\text{ReLU}}$ | 98.10 | 97.83 | 94.86 | 95.05 | $10_{30}$ | 28 | 3.9K |
| QRNN$_{\text{LeakyReLU}}$ | 98.01 | 97.96 | 95.13 | 94.86 | $10_{30}$ | 28 | 3.9K |
| QRNN$_{\text{GELU}}$ | 98.17 | **98.03** | 95.38 | **95.58** | $10_{30}$ | 28 | 3.9K |
| QRNN$_{\text{Linear}}$ | 97.06 | 96.80 | 94.94 | 94.13 | $10_{30}$ | 28 | 3.9K |
| QRNN$_{\text{Linear}}$ | 94.31 | 93.87 | 91.10 | 90.55 | $5_{15}$ | 28 | 1.3K |
| QRNN$^†$ | — | 96.70 | — | — | $q = 13$ | 1 | 3.1K |
| RNN | 97.42 | 97.28 | 95.16 | 94.28 | 50 | 28 | 3.9K |
| LSTM | 97.61 | 97.44 | 94.92 | 93.93 | 20 | 28 | 4K |
| scoRNN | 96.68 | 95.62 | 94.50 | 92.37 | 50 | 28 | 3.9K |

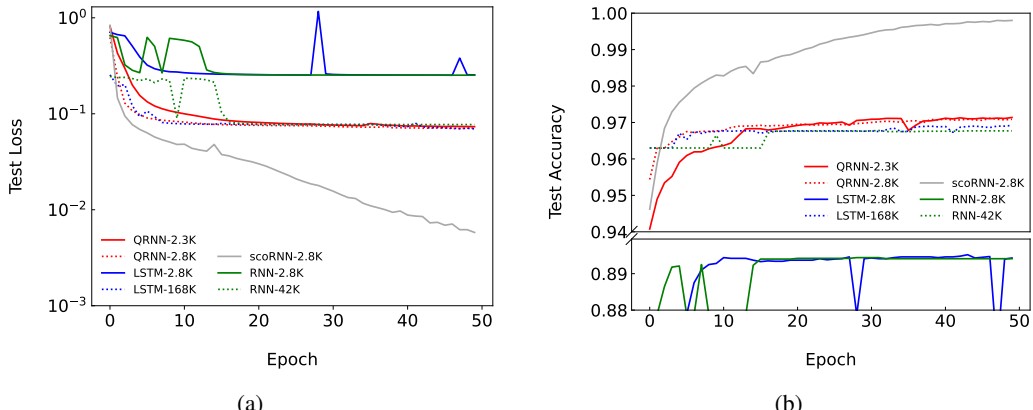

(a)                                           (b)

Figure 2: Test loss (a) and accuracy (b) for copying memory with $T = 200$.

### 4.3 COPYING MEMORY

The copying memory problem tests a model's ability to retain and recall information over long sequences (Hochreiter and Schmidhuber, 1997; Arjovsky et al., 2016). Each input sequence has $T + 20$ tokens, where the first $k = 10$ are random digits from 1 to 8 ($n_{\text{classes}}$), followed by zeros, and the last 11 ($k + 1$) positions are filled with the digit '9' with the first '9' acting as a delimiter. The model must learn to detect the delimiter and recall the original digits right after it in the output sequence. We randomly generated 5K training and 1K test samples with $T = 200$ (for training efficiency of QRNNs). A random guess baseline yields a loss of $\frac{k \cdot \log(n_{\text{classes}} - 1)}{T + 2k} \approx 0.095$, reflecting the expected cross-entropy when choosing uniformly from incorrect digits. On this task, QRNN-2.3K matches LSTM-168K (loss 0.07, accuracy 97%) and outperforms LSTM-2.8K (loss 0.25, accuracy 89.4%). scoRNN, specialized for this task, achieves near-perfect results, highlighting a performance gap between general-purpose and tailored models.

### 4.4 WORD-LEVEL LANGUAGE MODELING

The PTB dataset (Mikolov et al., 2011) consists of 929K training tokens, 73K validation tokens, and 82K test tokens. As is standard, we use a vocabulary size of 10K, converting OOV tokens to UNK. We

Table 3: PTB word-level language modeling (PPL). Qubit count $q$, total readouts $m$; or hidden state size $h$ (for RNN and LSTM only); embedding dimension $e$; parameter count $p$.

| Model | Val | Test | $q_m \vee h$ | $e$ | $p$ |
|---|---|---|---|---|---|
| QRNN$_{\text{ReLU}}$ | 131.81 | 126.69 | $14_{42}$ | 512 | 131K |
| QRNN$_{\text{LeakyReLU}}$ | 131.41 | 126.58 | $14_{42}$ | 512 | 131K |
| QRNN$_{\text{GELU}}$ | 136.62 | 131.07 | $14_{42}$ | 512 | 131K |
| QRNN$_{\text{LeakyReLU}}$ | 135.00 | 130.35 | $10_{30}$ | 512 | 78K |
| QRNN$_{\text{LeakyReLU}}$ | 169.17 | 161.09 | $5_{15}$ | 512 | 39K |
| RNN | 151.96 | 139.13 | 256 | 256 | 131K |
| LSTM | 124.22 | **120.30** | 128 | 128 | 132K |

Table 4: Multi30K German-to-English translation (BLEU). Qubit count $q$, total readouts $m$; or hidden state size $h$ (for RNN and LSTM only); embedding dimension $e$; parameter count $p$.

| Model | Val | Test | $q_m \vee h$ | $e$ | $p$ |
|---|---|---|---|---|---|
| QRNN$_{\text{GLU}}$ | 31.08 | 31.92 | $13_{39}$ | 512 | 412K |
| QRNN$_{\text{LeakyReLU}}$ | 29.22 | 28.99 | $13_{39}$ | 512 | 359K |
| QRNN$_{\text{GELU}}$ | 29.95 | 29.14 | $13_{39}$ | 512 | 359K |
| QRNN$_{\text{GLU}}$ | 30.16 | 31.51 | $10_{30}$ | 512 | 378K |
| QRNN$_{\text{GLU}}$ | 27.63 | 29.66 | $5_{15}$ | 512 | 320K |
| RNN | 29.50 | 29.50 | 512 | 293 | 412K |
| LSTM | 32.10 | 32.00 | 256 | 145 | 412K |
| scoRNN | 32.70 | **33.60** | 512 | 293 | 412K |

tested scoRNN on this task, but it did not converge to a good solution. The LSTM achieved the best result, with 120.30 perplexity (PPL), followed closely by QRNN$_{\text{LeakyReLU}}$ at 126.58.

## 4.5 MACHINE TRANSLATION

Soft attentions can be implemented using various formulations, such as additive attention or dot-product attention (Luong et al., 2015), but they share the same core principle: at each decoder timestep, compute a similarity score between the current decoder state and each encoder state, normalize these scores via a softmax, and form a context vector by summation, which is then combined with the decoder's hidden state to generate the next output token.

The attention mechanism implemented here follows the additive attention of Bahdanau et al. (2015). At each decoding step, the decoder hidden state is concatenated with encoder outputs, passed through a tanh activation followed by a linear projection to compute alignment scores. A softmax then normalizes these scores into attention weights, with masking applied to exclude padded positions.

We applied the model to Multi30k German-to-English translation (Elliott et al., 2016), with vocabulary sizes of 19.2K for German and 10.8K for English, and an average of 11 tokens per sentence in both languages. The training set contains 29K sentence pairs, with 1K each for validation and testing.

Results in Table 4 show that QRNN$_{\text{GLU}}$ with 13 qubits closely matches the LSTM, followed by QRNN$_{\text{GLU}}$ with 10 qubits. For the QRNN, it is somewhat surprising that intermediate readouts can still support mechanisms like soft attention, since these readouts capture only partial projections of the quantum state rather than the full hidden state. This suggests that, despite mid-circuit readouts, sufficient information is retained and propagated across timesteps. We qualitatively interpret the learned soft alignments on a few examples where the translations required non-trivial linguistic interpretations in Appendix A.

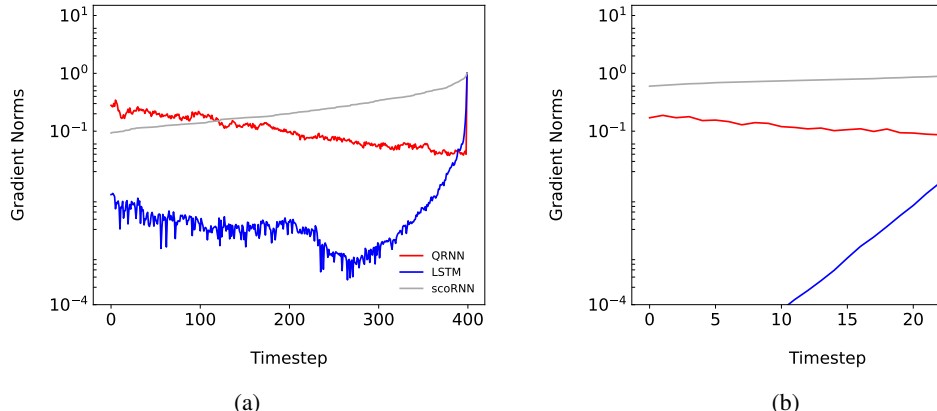

Figure 3: Normalized per-timestep gradient norms $\|\partial\mathcal{L}/\partial\mathbf{h}_t\|_2$, averaged over one mini-batch containing samples of identical $T$ (batch size = 16). Curves are normalized by the final timestep ($t = T$) gradient to compare decay shape; higher gradient values closer to $T = 0$ indicate less vanishing. (a) IMDB, $T = 400$. (b) pMNIST, $T = 28$.

### 4.6 Hidden State Gradients and Visualization

We measure per-timestep gradient norms on IMDB ($T = 400$) and pMNIST ($T = 28$) by retaining gradients on the per-timestep readouts (QRNN) and hidden states (LSTM) from saved checkpoints and computing $\|\partial\mathcal{L}/\partial\mathbf{h}_t\|_2$. Gradients are averaged across samples in a mini-batch and normalized by the last-step norm $\|\partial\mathcal{L}/\partial\mathbf{h}_T\|_2$ to compare decay shape.

As shown in Fig. 3, the QRNN curves remain consistently above the LSTM on both IMDB and pMNIST, indicating less vanishing through time toward the start of the sequences. All curves start with 1.0 at $t = T$ (normalization), but the relative elevation of the QRNN curve at earlier timesteps demonstrates more stable gradient propagation. The LSTM gradient norm decays rapidly, collapsing below $10^{-4}$ on the relatively short pMNIST sequences.

We visualize the per-timestep state trajectories for a single pMNIST test example in Appendix B.

## 5 Discussion and Conclusion

Different quantum hardware platforms currently require distinct control stacks, and architectural choices do not translate one-to-one across devices, with factors such as native gate sets, qubit connectivity, and the implementation of mid-circuit readout all affecting the realization of a given circuit. The aim here is not to prescribe a hardware roadmap but to analyze a hardware-realistic base case under idealized classical simulation to study the empirical properties of the architecture, where we model mid-circuit observations via expectation-value readouts.

As more efficient and scalable toolchains become available (e.g., future multi-GPU toolkits based on `cuQuantum` (Bayraktar et al., 2023)), we anticipate more faithful simulations via ancilla-mediated schemes in which auxiliary qubits are entangled with the main circuit, measured, and reset as needed while the recurrent memory remains coherent. This aligns with mid-circuit measure-and-reset operations already supported on several platforms (DeCross et al., 2022; Lis et al., 2023; Norcia et al., 2023).

This paper bridges quantum operations and recurrent learning by introducing a new hybrid QRNN whose recurrent core is implemented as a PQC steered by a classical controller. The unitary dynamics preserve norms, promoting stable gradient propagation; mid-circuit, per-timestep readouts inject task adaptability; and the classical controller supplies the nonlinearity and feedback for expressiveness. As techniques improve (Abbas et al., 2023) and quantum hardware matures, the architecture provides a path toward hardware-realistic quantum models for sequential learning.

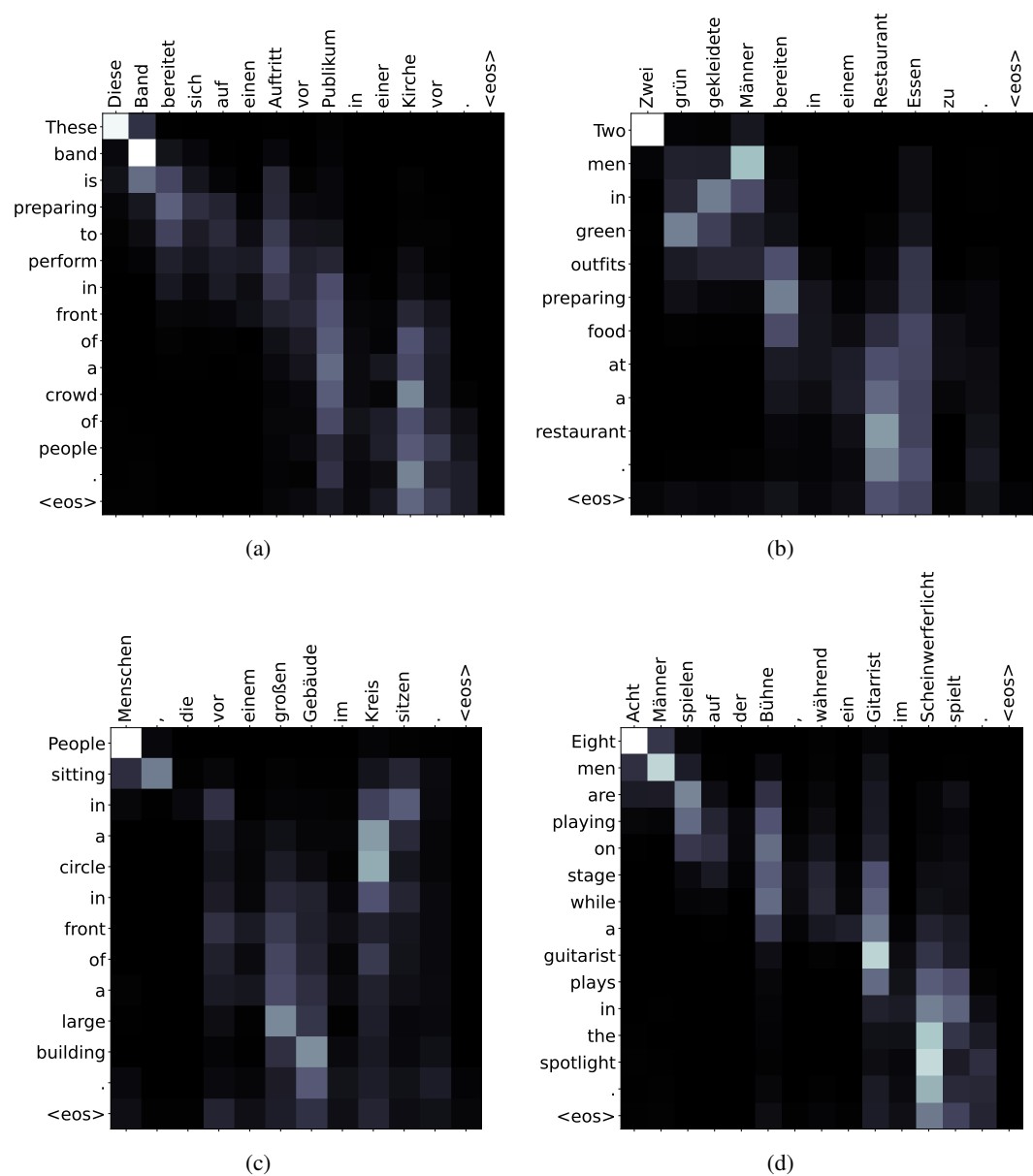

Figure 4: Soft attention alignments produced by the QRNN encoder-decoder model.

## A  ATTENTION ALIGNMENTS

To qualitatively analyze the model's learned soft attention alignments we selected four sentences from test set and interpreted the hybrid model translations and alignments (Fig. 4).

We observe that the hybrid model can manage spatial and syntactic shifts while capturing clause-level structure and semantics through its readout-based hidden states and soft attention as well as the LSTM baseline. It is evident that the model handles **compound verb constructions** and **semantic expansion**, in sentences like *"Diese Band bereitet sich auf einen Auftritt vor Publikum in einer Kirche vor"* (Fig. 4a) and *"Zwei grün gekleidete Männer bereiten in einem Restaurant Essen zu"* (Fig. 4b), where German separable verbs—*"bereitet … vor"* and *"bereiten … zu"*—are correctly reconstructed into the English verb phrases *"is preparing to perform"* and *"preparing"*, respectively. The soft attention allowed the model to attend across non-contiguous source tokens, enabling reassembly of

verb phrases. Additionally, lexical expansions such as *"Publikum"* → *"a crowd of people"* (Fig. 4a) and *"gekleidete Männer"* → *"men in green outfits"* (Fig. 4b) demonstrate contextually appropriate semantic elaboration beyond literal translation.

The model also displays **syntactic reordering** and **clause realignment**, necessitated by divergences between German and English word order. This is shown in both *"Diese Band . . . vor Publikum . . . vor"* and (Fig. 4a) *"Menschen, die vor einem großen Gebäude im Kreis sitzen"* (Fig. 4c). In the former, German's verb-final structure is reorganized into a mid-sentence English verb phrase, while handling nested prepositional phrases. In the latter, the relative clause *"die . . . sitzen"* is compressed into the participial phrase *"sitting"*, dropping auxiliaries and pronouns to better fit English syntactic norms. Similarly, the location and positional phrases *"im Kreis"* and *"vor einem großen Gebäude"* are reordered into *"in a circle in front of a large building"*

Lastly, for **multi-clause coordination**, **tense adaptation**, and **long-range dependency tracking**, as seen in *"Acht Männer spielen auf der Bühne, während ein Gitarrist im Scheinwerferlicht spielt"* (Fig. 4d). The model successfully disentangles two coordinated clauses and renders them with the correct English conjunction *"while"*, while adjusting verb forms from German's uniform *"spielen"* to *"are playing"* and *"plays"*, based on subject plurality. Finally, this ability to flexibly adapt clause boundaries and maintain coherence is also reflected in the *"Menschen . . . im Kreis sitzen"* example (Fig. 4c), where the model tracks relative clause dependencies and maps them onto compact English constructions.

## B HIDDEN STATE VISUALIZATION

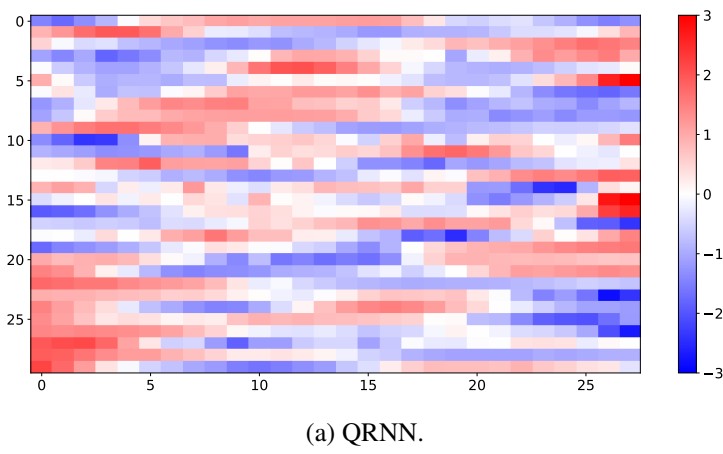

(a) QRNN.

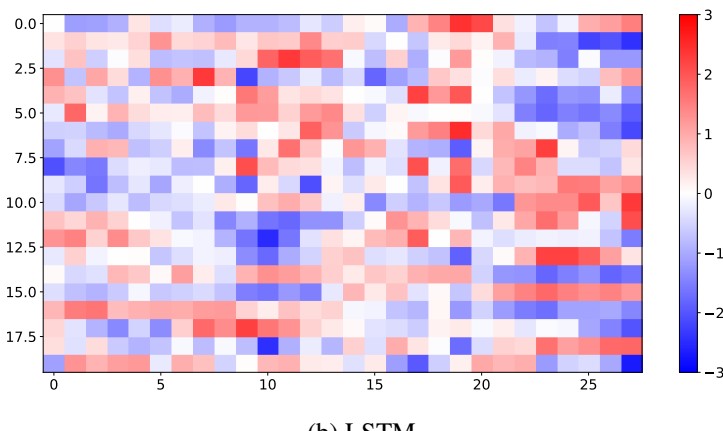

(b) LSTM.

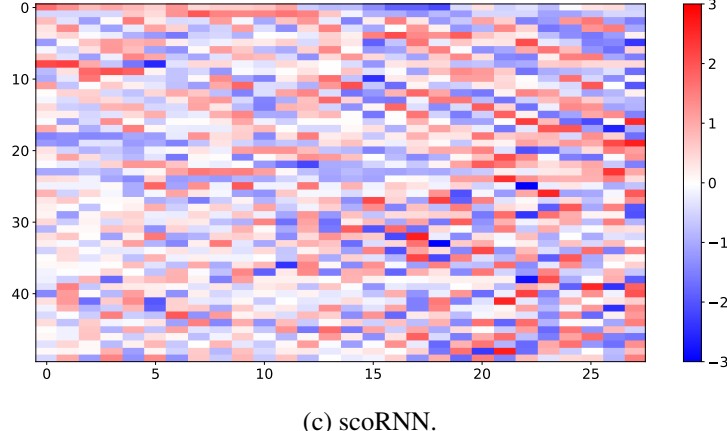

(c) scoRNN.

The above plots are obtained on a pMNIST sample. All models are those obtained the highest accuracies on the test set. For comparability across models with different dynamic ranges, each unit's trajectory is z-scored over time (per unit) and displayed using the same color scale (clipped to $[-3, 3]$). Qualitatively, the QRNN exhibits more coherent, banded temporal structure (units with sustained positive/negative excursions across multiple timesteps), whereas the LSTM and scoRNN activations appear more spatially fragmented and locally varying.

## C  EXPERIMENTAL SETTINGS AND TEST ACCURACY STATISTICS ACROSS RUNS

For RNN language modeling, we use the code from `https://github.com/yandex/faster-rnnlm`, and hyperparameters following the results and recommended settings in the documentation as below:

```
faster-rnnlm/rnnlm -rnnlm rnn.sig.256.1 -train ~/data/lm/ptb.train.txt
--valid ~/data/lm/ptb.valid.txt --hidden 256 --hidden-type sigmoid
--nce 50 --alpha 0.1 --bptt-skip 8 --bptt 35
```

For LSTM language modeling, we use the code from Zaremba et al. (2015). The hyperparameters are listed below:

```
batch_size=64, dropout=0.5, factor=1.0, factor_epoch=6, hidden_size=128,
layer_num=1, learning_rate=0.001, lstm_type='pytorch', max_grad_norm=5,
opt='adam', seq_length=35, total_epochs=49, winit=0.05
```

Table 5: QRNN Hyperparameters: batch size $b$, dropout rate $d$; embedding initialization $e_{init}$.

| Task | $b$ | $d$ | $e_{init}$ | $max\_grad\_norm$ |
|------|-----|-----|-----------|-------------------|
| IMDB | 200 | 0.25 | Xavier Uniform | 5 |
| MNIST | 200 | – | – | 5 |
| Copying (LeakyReLU) | 50 | – | – | 5 |
| PTB | 64 | 0.5 | Xavier Uniform | 5 |
| Multi30K | 64 | 0.25 | Xavier Uniform | 7 |

Table 6: RNN and LSTM Hyperparameters: batch size $b$, embedding dropout rate $d_e$; final output hidden state dropout rate $d_o$; embedding initialization $e_{init}$. The Adam hyperparameters are the same as the QRNN.

| Task | $b$ | $d_e$ | $d_o$ | $e_{init}$ | $max\_grad\_norm$ |
|------|-----|-------|-------|-----------|-------------------|
| IMDB | 100 | 0.2 | 0.2 | $\mathcal{N}(0,1)$ | 5 |
| MNIST | 200 | – | 0.2 | – | 1 |
| Copying | 50 | – | – | – | – |
| Multi30K | 64 | 0.25 | – | Xavier Uniform | 5 |

Table 7: scoRNN Hyperparameters: batch size $b$, embedding dropout rate $d_e$; final output hidden state dropout rate $d_o$; embedding initialization $e_{init}$. On MT, we use an internal PyTorch implementation of scoRNN which uses the same optimizer settings as our other models as described in the experimental settings. On other tasks, the hyperparameters are from the scoRNN paper's code including the optimizer hyperparamters.

| Task | $b$ | $d_e$ | $n_{neg\_ones}$ | $e_{init}$ |
|------|-----|-------|-----------------|-----------|
| IMDB | 64 | 0.2 | $h/10$ | Uniform $(-1,1)$ |
| MNIST | 50 | – | $h/10$, $h/2$ (permuted) | – |
| Copying | 50 | – | $h/2$ | – |
| Multi30K | 64 | 0.5 | $h/10$ | Xavier Uniform |

Table 8: Accuracy statistics on MNIST and pMNIST test sets across 50 runs for each nonlinearity variant. Qubit count $q$ and total readouts $m$; embedding dimension $e$; parameter count $p$. scoRNN is reported over 10 runs.

| Model | MNIST | | | pMNIST | | | $q_m \vee h$ | $e$ | $p$ |
|-------|-------|-------|-------|--------|-------|-------|--------------|-----|-----|
| | $min$ | $max$ | $\mu$ | $min$ | $max$ | $\mu$ | | | |
| QRNN$_{\text{ReLU}}$ | 97.51 | 98.25 | 97.84 | 94.33 | 95.31 | 94.83 | $10_{30}$ | 28 | 3.9K |
| QRNN$_{\text{LeakyReLU}}$ | 97.42 | 98.15 | 97.88 | 94.33 | 95.38 | 94.80 | $10_{30}$ | 28 | 3.9K |
| QRNN$_{\text{GELU}}$ | 97.62 | 98.22 | 97.96 | 94.72 | 95.58 | 95.12 | $10_{30}$ | 28 | 3.9K |
| RNN | 96.36 | 97.32 | 96.92 | 93.66 | 94.53 | 94.11 | 50 | 28 | 3.9K |
| LSTM | 96.62 | 97.63 | 97.23 | 93.35 | 94.09 | 93.74 | 20 | 28 | 4K |
| scoRNN | 94.82 | 95.92 | 95.26 | 92.15 | 92.83 | 92.42 | 50 | 28 | 3.9K |

Table 9: PPL on PTB test set across 5 runs for each nonlinearity variant. Qubit count $q$, total readouts $m$; embedding dimension $e$; parameter count $p$. With the `faster-rnnlm` toolkit for RNN, we obtained the same results on multiple runs as Table 3.

| Model | $min$ | $max$ | $\mu$ | $q_m \vee h$ | $e$ | $p$ |
|-------|-------|-------|-------|--------------|-----|-----|
| QRNN$_{\text{LeakyReLU}}$ | 126.53 | 128.82 | 127.77 | $14_{42}$ | 512 | 131K |
| LSTM | 119.83 | 121.99 | 120.57 | 128 | 128 | 132K |

Table 10: BLEU evaluations on the Multi30K German to English test set across 20 runs for each nonlinearity variant. Qubit count $q$, total readouts $m$; embedding dimension $e$; parameter count $p$.

| Model | $min$ | $max$ | $\mu$ | $q_m \vee h$ | $e$ | $p$ |
|---|---|---|---|---|---|---|
| $\text{QRNN}_{\text{GLU}}$ | 19.83 | 31.92 | 27.88 | $13_{39}$ | 512 | 412K |
| $\text{QRNN}_{\text{LeakyReLU}}$ | 24.52 | 29.87 | 28.55 | $13_{39}$ | 512 | 359K |
| $\text{QRNN}_{\text{GELU}}$ | 25.71 | 30.29 | 29.09 | $13_{39}$ | 512 | 359K |
| RNN | 27.60 | 29.90 | 29.16 | 512 | 293 | 412K |
| LSTM | 32.00 | 33.10 | 32.42 | 256 | 145 | 412K |
| scoRNN | 29.50 | 33.60 | 32.34 | 256 | 293 | 412K |

Table 11: Accuracy statistics on IMDB test set across 100 runs for each nonlinearity variant. Qubit count $q$, total readouts $m$; embedding dimension $e$; parameter count $p$. RNN and LSTM are reported over 50 runs. scoRNN is reported over 5 runs.

| Model | $min$ | $max$ | $\mu$ | $min^*$ | $\mu^*$ | $q_m \vee h$ | $e$ | $p$ |
|---|---|---|---|---|---|---|---|---|
| $\text{QRNN}_{\text{ReLU}}$ | 49.55 | 85.96 | 71.18 | 71.74 | 83.11 | $8_{24}$ | 100 | 5K |
| $\text{QRNN}_{\text{LeakyReLU}}$ | 49.63 | 87.00 | 70.23 | 75.77 | 83.44 | $8_{24}$ | 100 | 5K |
| $\text{QRNN}_{\text{GELU}}$ | 49.98 | 86.38 | 77.18 | 70.39 | 83.75 | $8_{24}$ | 100 | 5K |
| RNN | 85.24 | 87.19 | – | – | 86.39 | 50 | 50 | 5K |
| LSTM | 86.28 | 88.00 | – | – | 86.99 | 25 | 25 | 5.2K |
| scoRNN | 85.57 | 86.48 | – | – | 86.19 | 50 | 50 | 5K |

Among all tasks, IMDB showed the greatest variability in QRNN performance across random seeds in development. This behavior may align with known sensitivities in training variational PQCs (Grant et al., 2019). We therefore also report stats where we remove failed runs ($< 70\%$ accuracy, well below simple baselines such as BoW), indicated by $*$. For the three nonlinearities 40, 42 and 25 failed runs were observed each. The results also indicate that GELU nonlinearity reduces the sensitivity compared with the other two.

While parametrized quantum circuits (PQCs) can suffer from vanishing gradients in deep or wide settings due to the barren plateau phenomenon (McClean et al., 2018), there is no general impossibility theorem that barren plateaus must occur in all parametrized quantum circuits; their presence and severity are known to depend on the ansatz, cost function, initialization, training strategy, and noise, and remain an empirical matter at practical scales. Several studies provide insights into how it arises or design principles that prevent or mitigate plateaus (Cerezo et al., 2019; Grant et al., 2019; Patti et al., 2021; Sack et al., 2022). These results indicate that barren plateaus are not inevitable, and that careful design yields a tractable and stable training landscape in practice. In particular, some architectures such as quantum convolutional neural networks avoid barren plateaus by construction (Pesah et al., 2021), which supports the view that appropriate architectural choices can produce stable and trainable quantum models.

## D   QUANTUM STATES AND SUPERPOSITION

Unlike a classical bit, a qubit exists in a *superposition* of the states 0 and 1 in a two-dimensional complex Hilbert space: $|\psi\rangle = \alpha|0\rangle + \beta|1\rangle = [\alpha \quad \beta]^T \in \mathbb{C}^2$ and $|0\rangle = [1 \quad 0]^T$ and $|1\rangle = [0 \quad 1]^T$ are elements of the *computational basis* for the Hilbert space. The coefficients $\alpha$ and $\beta$ are complex numbers referred to as the *amplitudes* that satisfy $|\alpha|^2 + |\beta|^2 = 1$. For a state $|\psi\rangle = \alpha|0\rangle + \beta|1\rangle$, the probability of obtaining $|0\rangle$ is $|\alpha|^2$, and the probability of obtaining $|1\rangle$ is $|\beta|^2$.

## E  PQC TEMPLATE

We have chosen the PQC template based on the benchmarking study in Sim et al. (2019), which evaluates 19 different parametrized quantum circuits (PQCs) up to depth 5 (i.e., the base circuit repeated up to five times and used a single PQC). Each PQC is assessed using two key metrics: expressibility and entangling capability. The architecture referred to as ansatz-14 in Sim et al. (2019) which we use here in a single layer configuration was shown to score highly on both. This gives a good balance of simulation cost and "goodness" of the PQC.

Expressibility is quantified by comparing the distribution of pairwise fidelities between states generated by the PQC to the theoretical fidelity distribution of Haar-random states, which represent uniform randomness over the composite Hilbert space (the tensor product of individual qubit spaces). Instead of generating Haar-random states directly, the method in (Sim et al., 2019) uses the analytical form of the Haar fidelity distribution as a reference. PQC output states are obtained by sampling random parameters, and their pairwise fidelities are used to construct an empirical distribution. The KL divergence between this empirical distribution and the Haar reference provides a scalar expressibility score, with lower values indicating greater expressiveness.

There has also been recent work on Quantum Deep Equilibrium Models (QDEQs), which use deep equilibrium ideas to train PQC-based quantum models with effectively lower circuit depth and fewer parameters (Schleich et al., 2024). Orthogonal to the above, this suggests a promising path to shallower circuits while maintaining or enhancing expressivity.

## F  COPYING TASK RESULTS SUMMARY

Table 12: Copying task summary results of Fig. 2 for models with the same number of parameters.

| Model | Acc. | Loss | $q_m \vee h$ | $e$ | $p$ |
|---|---|---|---|---|---|
| QRNN$_{\text{LeakyReLU}}$ | 97.09 | 0.0706 | $8_{24}$ | 10 | 2.8K |
| RNN | 89.41 | 0.2524 | 48 | 10 | 2.8K |
| LSTM | 89.43 | 0.2543 | 22 | 10 | 2.8K |
| LSTM | 96.92 | 0.0697 | 200 | 10 | 169.6K |
| scoRNN | **99.80** | **0.0058** | 48 | 10 | 2.8K |

## G  WALL-CLOCK TRAINING TIME COMPARISON

Table 13: Per-epoch wall-clock training time for QRNN and LSTM, using the best-performing model configuration for each task.

| Task | QRNN ($\sim$ mins) | LSTM ($\sim$ secs) |
|---|---|---|
| IMDB | 8 | 10 |
| MNIST | 4 | 7 |
| Copying | 9 | 1 |
| PTB | 36 | 5 |
| Multi30K | 31 | 38 |

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
