# OpenReview forum: "Hybrid Quantum-Classical Recurrent Neural Networks"
_ICLR.cc/2026/Conference — Submitted to ICLR 2026_

### Official Review · Reviewer_FAnR · 2025-10-30

**Soundness:** 2
**Presentation:** 2
**Contribution:** 3
**Rating:** 4
**Confidence:** 3

**Summary:**

The paper presents a hybrid-classical quantum recurrent neural network. In the proposed architecture, the hidden state is the quantum state of a parametric quantum circuit (PQC). The parameters of the PQC, in turn, are controlled by a classical neural network that processes both the prior hidden state (or rather, a measurement taken from it) and the new input.
The paper provides an empirical study with extensive experiments on various datasets.

**Strengths:**

- Designing algorithms and architectures that allow to integrate quantum computing into machine learning is an important task. Thereby, the paper contributes to a relevant research area.
- The experiments are carried out and described (mostly) very thoroughly, providing important experimental insights on the performance of QRNN architectures in comparison to classical architectures.
- In the experimental study, the proposed architecture appears to perform well in comparison to classical architectures.
- Overall, the paper is well-written and clearly presented.

**Weaknesses:**

- Important experimental details on baselines are missing: the precise choices for number of hidden layers, activation functions, etc. for the classical baseline models (RNNs, LSTMs) are not reported. While Dropout is applied for the QRNNs, it is not stated whether it is also used for the classical baseline models. Moreover, it remains unclear how exactly the network sizes / total number of parameters were chosen for the different experiments. This makes it impossible to reproduce experimental results and undermines the main claims of the paper.

- Claimed outperformance not supported by experiments:  the paper states that QRNNs outperform standard LSTMs/RNNs on four of six tasks. However, this is not fully supported by the experiments: only in two of the experiments QRNNs appear to really obtain a significant improvement (Table 2). For the remaining experiments, this does not seem to be the case. For the first task (reported in Table 1), the difference between the RNN performance and the best QRNN is very minor, while the RNN uses fewer parameters (only 5K instead of 5.2K). For the task in Section 4.3, no table is provided, but the text states that LSTMs and QRNNs perform equally well. For the remaining two tasks LSTMs outperform QRNNs. This would mean that only in 2/6 tasks QRNNs outperform classical methods.
Moreover, to allow for a full comparison of the considered architectures, it would be crucial to also report runtimes.

- Only classical baselines used: several other recurrent quantum neural networks architectures have been proposed (Ubale et al. 2025, Yu et al. 2024a, etc. as discussed in Section 2), but the paper does not compare performance to any of them. This would be important for justifying why yet another QRNN architecture should be considered. In particular, this is important because in comparison to these works the novelty of the architecture is incremental (adding non-linearity to the circuits from Li et al. 2023 and Siemaszko et al. 2023)

- No code is available with the paper, which limits reproducibility of the experimental results.

- Missing details on the methodology: the model architecture is never spellt out in full detail, which makes it not possible to reproduce the results. Figure 1(a) is stated as an example, but the paper does not state how exactly the unitary operator U looks like in general and which gates are used exactly. As for measurement, lines 260-262 state that "measurement outcomes are combined to form $z_t$, but does not state how exactly these measurements are combined.

- Impact of measurement noise: Related to this point, it appears that the measurement operation in (3) would be prone to noise. This needs to be thoroughly addressed in the paper, as it may have a major impact on the performance. How is this addressed in the current implementation? If it is currently neglected, it would be highly important to assess its impact on the performance.

- The discussion on norm preservation is misleading: only the norms of the quantum states are preserved, but this does not have any impact on or relation to the norms of the inputs.

**Questions:**

- Can you provide full experimental details regarding the baselines (use of dropout, activation functions, etc., see first point above)?
- How sensitive are the baselines with respect to choices of activation functions?
- According to which procedure did you select network sizes / hyperparameters / activation functions for the classical baselines?
- Can you add a table with results also for Section 4.3?
- What happens in Table 1 if you increase the number of RNN parameters to 5.2K?
- Can you provide insights on the use of alternative, previously proposed QRNN architectures?
- Can you provide code for your implementation?
- Can you write out in detail the model architecture: how exactly is U defined and how does the measurement operator look like.
- Can you assess the impact of measurement noise on the results?
- The paper states "To preserve coherence across timesteps, we simulate mid-circuit measurements, allowing recurrent structure without collapsing the full quantum state, retaining the quantum memory throughout the sequence." It appears that this would create problems on real quantum hardware  - can you comment on this?

---

> ### Author Response · Authors · 2025-11-27
> **Re: Weakness 1 -- 2**
>
> We sincerely thank the reviewer for comments, and appreciate the opportunity to clarify.
>
> > Important experimental details on baselines are missing: the precise choices for number of hidden layers, activation functions, etc. for the classical baseline models (RNNs, LSTMs) are not reported. While Dropout is applied for the QRNNs, it is not stated whether it is also used for the classical baseline models.  Moreover, it remains unclear how exactly the network sizes / total number of parameters were chosen for the different experiments. This makes it impossible to reproduce experimental results and undermines the main claims of the paper.
>
> All hyperparameter are tuned on the validation splits and settings for all baselines which we intended to add are now in the Appendix C of the uploaded rebuttal revision. We briefly describe them below:
>
> - The main hyperparamters which are the ones associated with Adam, and those were actually tuned with the LSTM models. For all QRNNs/RNN/LSTMs/scoRNNs (for MT with our own PyTorch impl.) we use the same set of settings which are $(lr = 1\times10^{-3},\ \lambda = 1\times10^{-4},\ \text{and}\ \epsilon = 1\times10^{-10})$, which we reported in Sec. 4.
>
> - scoRNN uses RMSProp and associated hyperparameters of it from the released TensorFlow codel. We reimplemented a PyTorch version for the MT task, which uses the same Adam as other models.
>
> - The number of hidden layers is always 1 for all models. RNNs all use ReLU (except for language modelling, where we use this toolkit https://github.com/yandex/faster-rnnlm and the settings reported in their documentation which showed Sigmoid resulted in better accuracy than ReLU for language modelling in combination of other tricks such as Noise Contrastive Estimation); LSTMs are PyTorch defaults.
>
> - Dropouts are used for all except (p)MNIST and Copying where the inputs are fixed. For IMDB and (p)MNIST RNNs and LSTMs use dropout at the output layer in addition to embedding dropout. QRNNs use embedding dropout only.
>
> - The baseline total number of parameters (hidden layer size, embedding size) are chosen such that the total parameter count matches the QRNNs.
>
> > How sensitive are the baselines with respect to choices of activation functions?
>
> RNNs all use ReLU. LSTMs use PyTorch defaults. scoRNN uses its own ModReLU activation as reported in the paper, suitable for it's own orthogonal parametrization.
>
> > Claimed outperformance not supported by experiments: the paper states that QRNNs outperform standard LSTMs/RNNs on four of six tasks. However, this is not fully supported by the experiments: only in two of the experiments QRNNs appear to really obtain a significant improvement (Table 2). For the remaining experiments, this does not seem to be the case. For the first task (reported in Table 1), the difference between the RNN performance and the best QRNN is very minor, while the RNN uses fewer parameters (only 5K instead of 5.2K).
>
> We'd like to clarify a few things here. In the submitted version, the parameter count column of the IMDB experiments had the counts for LSTM and QRNN swapped, correct count for LSTM is 5.2K while, both RNN and QRNN should be 5K. We have updated table in the rebuttal revision. RNN therefore did not have less number of parameters than QRNN. Our overall goal is also not to maximize performance gains, and all results are competitive with the classical baselines. We have removed any references to "outperforming" with "competitive" in the updated rebuttal revision. We would like to clarify that we do not intend this model to be run on classical hardware as a competitive alternative to current classical architectures, nor do we position it as a near-term solution on NISQ devices. The goal of this work is not to outperform classical models under today’s conditions, but to establish viability of a feedback-driven quantum recurrent architecture. We have added the following to the uploaded rebuttal revision:
>
>
> > For the task in Section 4.3, no table is provided, but the text states that LSTMs and QRNNs perform equally well. For the remaining two tasks LSTMs outperform QRNNs. This would mean that only in 2/6 tasks QRNNs outperform classical methods. Moreover, to allow for a full comparison of the considered architectures, it would be crucial to also report runtimes.
>
> We follow the convention in previous papers (such as [1] and [2]) for the copying task where learning and accuracy curves are used, mainly to also show rate of convergence in addition to final results. However, we've now added a table of results summary in the uploaded rebuttal revision in Appendix  F.
>
> [1] Unitary RNN https://arxiv.org/pdf/1511.06464
> [2] scoRNN https://arxiv.org/abs/1707.09520

---

> > ### Author Response · Authors · 2025-11-27
> > **Re: Weakness 3 -- 4**
> >
> > > Only classical baselines used: several other recurrent quantum neural networks architectures have been proposed (Ubale et al. 2025, Yu et al. 2024a, etc. as discussed in Section 2), but the paper does not compare performance to any of them. This would be important for justifying why yet another QRNN architecture should be considered.
> >
> > Ubale et al., 2025 and Yu et al, 2024 are fundamentally different architectures to ours.
> > They replace dense layers in the LSTM gates with PQCs. However, all memory and recurrence remain entirely classical, governed by standard hidden and cell state updates. These architectures are best viewed as classical LSTMs augmented with auxiliary PQCs, rather than quantum recurrent models. Ubale et al., 2025 only investigate fraud detection while Yu et al., 2024 focuses on low-resource language text classification.
> >
> > A detailed empirical comparison would require re-implementing their models and/or adapting them to our full suite of benchmarks (including copying memory and sequence-to-sequence translation), which we view as an orthogonal undertaking. In this paper, we focus on comparing against classical recurrent baselines while providing the first empirical study of a genuinely quantum-recurrent architecture with mid-circuit feedback. To our knowledge, our QRNN is the first model (RNN or otherwise) to be shown on the suite of tasks we considered here.
> >
> > >In particular, this is important because in comparison to these works the novelty of the architecture is incremental (adding non-linearity to the circuits from Li et al. 2023 and Siemaszko et al. 2023)
> >
> > The circuits we used are completely different from Li et al. 2023 and Siemaszko et al. 2023. In addition, our main contributions and novelty are:
> >
> > - A new hybrid recurrent mechanism that combines 1) coherent quantum memory with 2) nonlinear classical control and 3) mid-circuit readouts in a novel feedback scheme.
> >
> > - The first empirical demonstration of such a quantum model (RNN or otherwise) across realistic sequence-learning tasks (at the scaling limits of single-gpu simulations and limitations of current toolchains). We also demonstrate for the first time measurement readouts can be successfully used in soft attention sequence2sequence learning.
> >
> > - Our architecture provides a concrete, trainable recurrent mechanism identified in recent concurrent theory, linking universality results to an actual implementable design.
> >
> > These contributions establish the first conceptual motivation and the practical viability of a hybrid QRNN, aligned with a universality theory (https://openreview.net/forum?id=248ysaRatx).
> >
> > > No code available.
> >
> > Code will be made availabe and we are preparing the code and scripts for reproducing the experiments for release.

---

> > > ### Author Response · Authors · 2025-11-27
> > > **Re: Weakness 5 -- 7**
> > >
> > > > Missing details on the methodology: the model architecture is never spellt out in full detail, which makes it not possible to reproduce the results. Figure 1(a) is stated as an example, but the paper does not state how exactly the unitary operator U looks like in general and which gates are used exactly.
> > >
> > > We'd like to clarify that we first explicitly explained in the Caption of Fig. 1 the definition of the U:
> > >
> > > "The feedforward network $\mathcal{F}$ takes the combined vector $(z_{t−1} : x_t)$ and produces $|\theta_t|$ = *16* parameters per timestep that control the PQC structure from (a), denoted $U(\theta_t)$." (where (a) referes to Fig. 1(a).
> > >
> > > Then again, we explained in the intro (bottom of p. 2 bullets and again in Sec 3.2 line 234). We have also marked all the gates in the PQC with numbered subscripts from 1 to *16*.
> > >
> > >
> > > > As for measurement, lines 260-262 state that "measurement outcomes are combined to form, but does not state how exactly these measurements are combined.
> > >
> > > Thanks for raising this. We said on line 261 they are combined to form $z_t$. However, we've revised in the uploaded rebuttal version on lines 232 - 236 to make this more explicit that it's clear that they are combined by concatenation.
> > >
> > > > Impact of measurement noise: Related to this point, it appears that the measurement operation in (3) would be prone to noise. This needs to be thoroughly addressed in the paper, as it may have a major impact on the performance. How is this addressed in the current implementation? If it is currently neglected, it would be highly important to assess its impact on the performance.
> > >
> > > We agree that our experiments are purely in noiseless classical simulations, and we are explicit about this in the paper. This is deliberate rather than an oversight. Our goal here is not to claim an immediate NISQ-era (ie noisy) hardware advantage, but to establish that the feedback-driven quantum recurrent architecture is (i) well-posed, (ii) trainable with gradient methods, and (iii) competitive with strong classical RNN/LSTM baselines on broad-spectrum realistic sequence tasks, under noiseless simulations. In our view, this is a necessary first step before it is meaningful to discuss scaling on fault-tolerant hardware. If a model does not demonstrate viability in noiseless simulations, which provide the most favourable possible setting for learning, then adding hardware noise (at inference time only with the lack of scalable quantum backprop atm) can only degrade its behaviour further, and the experiments more convoluted, making it unclear how such an architecture could ever be made plausible on real devices.
> > >
> > > Further, introducing ad-hoc noise channels (with platform-dependent parameters) on top of our simulation would not tell us whether the architecture itself is learnable: such noise would be injected only *at inference*, not during training, and there is currently no practical quantum backpropagation method that would allow us to train the same model natively on hardware. As a result, noisy forward-only simulations would conflate architectural viability with arbitrary implementation details of a particular device, while still leaving the central question "can this feedback-driven QRNN be trained effectively", unanswered. We therefore, like many experimental studies, focus on the clean, ideal setting and treat noise as an orthogonal, future engineering problem once training-capable hardware is available.
> > >
> > > We have to also clarify that we do not intend the model to be run on classical computers to compete with current models or even on NISQ devices. We see this as a foundational first step to establish viability and we added to the last section: "[...] although hardware implementations for large-scale sequence modeling would require
> > > fault-tolerant devices capable of sustaining long coherent recurrences and real-time classical control."
> > >
> > > > The discussion on norm preservation is misleading: only the norms of the quantum states are preserved, but this does not have any impact on or relation to the norms of the inputs.
> > >
> > > Our intention is not to suggest that the QRNN preserves the norms of the inputs; rather, we refer specifically and explicitly throughout the paper to the norm of the recurrent hidden state and the associated gradient dynamics through time, as we show the in Sec 4.6. We have now also added hidden state visualisations in Appendix B. The main source of problematic BPTT gradients lies in the recurrent transition itself. Classical orthogonal/unitary RNNs likewise do not preserve input norms; instead, they preserve the norm of the hidden state, which stabilizes the recurrent Jacobian and leads to more well-behaved gradients through time.
> > >
> > > If we missed anything here, pls kindly let us know.

---

> ### Author Response · Authors · 2025-11-27
> **Re: Questions**
>
> We thank the reviewer again for their time and comments, we trust all questions except the following tw0 have been addressed above.
>
> > The paper states "To preserve coherence across timesteps, we simulate mid-circuit measurements, allowing recurrent structure without collapsing the full quantum state, retaining the quantum memory throughout the sequence." It appears that this would create problems on real quantum hardware - can you comment on this?
>
> We'd like to emphasize that we work in the noise-free "limit" and maintaining coherence of the quantum state across time; mathematically, the mid-circuit readouts and feedback are implemented via expectation readouts (we have clarified all places where we used "measurment" in the paper with expectation-value readouts). There are also two practical ways to relax this: 1) by the universality proof, we can do projective measurements and reinitialize at the next timestpe, and 2) as we suggested in the discussion section "via ancilla-mediated schemes in which auxiliary qubits are entangled with the main circuit, measured, and reset as needed while the recurrent memory remains coherent. This aligns with mid-circuit measure-and-reset operations already supported on several platforms (DeCross et al., 2022; Lis et al., 2023; Norcia et al., 2023), although hardware implementations for large-scale sequence modeling would require fault-tolerant devices capable of sustaining long coherent recurrences and real-time classical control."
>
> > Can you provide insights on the use of alternative, previously proposed QRNN architectures?
>
> (pls kindly refer to Section 2 Related Work of the paper and the following.)
>
> here are mainly two categories of "quantum" models for classical tasks: (1) quantum-inspired models (such as Li et al. referred to in the question), which are not intended to run on quantum hardware and remain classical despite having quantum-inspired components or terminology; and (2) quantum models with hybrid components for handling classical data, where the quantum parts are intended to be actual quantum operations (with the potential to benefit from actual quantum hardware, not just quantum-inspired).
>
> Specifically, the model of Li et al. is a complex-valued classical Transformer, which is strictly a classical model that replaces real numbers with complex numbers (type 1 above). In contrast, there are also quantum "Transformers" based on purely quantum primitives/operations that have been demonstrated to run (at inference) on quantum hardware (type 2 above), e.g., [0].
>
> For RNNs, for example [1, 2 etc.] and our work are all type 2 (and Li et al. (2023) and Siemaszko et al. (2023) discussed in the related work section among others), while [3,4 etc.] are type 1.
>
> [0] Quantum vision transformer (https://quantum-journal.org/papers/q-2024-02-22-1265/pdf/)
>
> [1] Bausch 2020, Recurrent quantum neural networks (https://proceedings.neurips.cc/paper/2020/hash/0ec96be397dd6d3cf2fecb4a2d627c1c-Abstract.html)
>
> [2] Nikoloska et al., Time-warping invariant quantum recurrent neural networks via quantum-classical adaptive gating
>
> [3] Chen et al. Quantum Long Short-Term Memory (https://arxiv.org/abs/2009.01783)
>
> [4] Ubale et al. Toward practical quantum machine learning: A novel hybrid quantum lstm for fraud detection (https: //arxiv.org/abs/2505.00137)
>
> For the rebuttal revision given space limitation, we will expand the related work section in later revisions to further clarify these distinctions.

---

### Official Review · Reviewer_vQW1 · 2025-10-30

**Soundness:** 3
**Presentation:** 3
**Contribution:** 2
**Rating:** 4
**Confidence:** 4

**Summary:**

This paper introduces a hybrid quantum-classical RNN architecture in which the recurrent core is implemented as a PQC. At each timestep, mid-circuit measurements from the previous quantum state are combined with classical embeddings of the current input through a feedforward network, which outputs the next set of PQC parameters. The quantum state acts as the hidden state, evolving unitarily in a high-dimensional Hilbert space. The authors argue that this design offers a principled way to integrate unitary evolutions into recurrent architectures while preserving state norm and enabling per-timestep readouts.
The model is evaluated in simulation (up to 14 qubits) on standard benchmarks including sentiment analysis, MNIST, permuted MNIST, copying memory, and machine translation. The authors report modest accuracy improvements over classical RNN baselines.

**Strengths:**

Novel architectural concept: The idea of embedding quantum circuits within RNN recurrence steps is original and conceptually interesting, especially for sequence modeling.
Comprehensive benchmarking: The authors evaluate the model across diverse tasks, including language modeling and translation, showing its general applicability.
Mathematical consistency: The recurrent evolution is unitary by construction, automatically ensuring norm preservation—an elegant contrast to the ad-hoc regularization required in classical RNNs.
Practical transparency: The paper reports training times, qubit counts, and parameter sizes for both classical and hybrid models, which is commendable.

The work presents an interesting blueprint for integrating parametrized quantum circuits into recurrent models, but the evidence for impact is limited. The modest performance improvements could stem from added representational capacity rather than genuine quantum effects. The unclear separation between encoding and learning blocks, combined with incomplete training details, makes replication and interpretation difficult.
Given that all experiments rely on classical simulation and incur high computational costs, the results do not substantiate the paper’s implicit suggestion that such architectures are practical or advantageous in the near term.

**Weaknesses:**

Ambiguity in circuit design: The paper does not clearly distinguish between data-encoding and trainable parts of the PQC. From Figure 1A, the first layer of RY gates seems to correspond to data encoding, but this is not explicitly stated. Without this separation, it is difficult to evaluate what portion of the model’s expressivity truly arises from quantum effects.

Modest empirical gains: Across tasks, improvements are small (e.g., +2\% accuracy in classification, BLEU 29.2 --> 31.9 for German–English translation). These gains do not convincingly justify the significant simulation cost or architectural complexity, especially when comparable improvements could be achieved by scaling classical models.

Limited discussion of simulation cost: Although the authors mention that training takes between 4–60 minutes per epoch (depending on task and qubit count), no wall-clock comparisons are provided against classical baselines. Given that all results are obtained on simulators rather than real quantum hardware, the practical feasibility and scalability of the method remain unclear.

Measurement and training details missing: The paper reports “24 measurements for 8 qubits” but does not specify whether this is per timestep or per sample, nor how parameter gradients were obtained. If the parameter-shift rule was used, the number of measurements would change, significantly impacting the total computational cost. Clarification is needed to assess efficiency and scalability.

Unclear goal between physics and simulation: It remains ambiguous whether the main objective is to demonstrate a computationally advantageous quantum model or to motivate unitary evolution as a conceptual analogue for classical nonlinearities. Without a clear experimental focus, the narrative risks oscillating between hardware-motivated and purely simulated reasoning.

Related work gap: Although several hybrid sequence models are cited, the omission of Quantum Deep Equilibrium Models is notable, as it directly addresses the encoding overhead and representational collapse issues that also affect this architecture.

While conceptually creative, the paper does not provide compelling empirical or theoretical evidence that hybrid quantum-classical RNNs outperform, generalize better, or offer unique interpretability compared to classical counterparts. The design lacks clarity regarding the role of data encoding and omits key implementation details needed to evaluate its feasibility on hardware. The work’s contribution lies more in architectural exploration than in demonstrated quantum advantage.

**Questions:**

In this type of architecture, reading and feeding back to the QCP could have a great cost; is it the case for this QML model too?

Are canonical initialization schemes (e.g., amplitude or angle encoding) used for classical input embeddings?

Is the reported “24 measurements for 8 qubits” the number of observables per timestep or per sample? For training, how was gradient estimation done (e.g., via parameter-shift rule)?

How do the runtime costs per epoch compare quantitatively against the classical RNN baseline (e.g., same hardware and dataset)?

Will replacing the PQC with an orthogonal matrix RNN test whether unitarity alone explains observed gains?

---

> ### Author Response · Authors · 2025-11-27
> **Re: Weakness 1**
>
> We would like to sincerely thank the reviewer for comments, and we appreciate the opportunity to make clarifications.
>
> > Ambiguity in circuit design: The paper does not clearly distinguish between data-encoding and trainable parts of the PQC. From Figure 1A, the first layer of RY gates seems to correspond to data encoding, but this is not explicitly stated. Without this separation, it is difficult to evaluate what portion of the model’s expressivity truly arises from quantum effects.
>
> We thank the reviewer for this helpful comment. In our architecture, we do not follow the common pattern of separating a fixed data-encoding unitary from a subsequent trainable PQC block. Instead, this is unified: the entire parametrized ansatz in Fig. 1(a) is controlled by the classical network $F$ and is re-programmed at every timestep. There is no subset of gates that serves as a non-trainable “data-encoding layer” acting only on $x_t$.   The gates in Fig. 1(a) are explicitly marked with subscripted numbers from 1 to 16 which correspond to the outputs of the feedforward net at each step, which reparemetrizes all the 16 gates of the PQC (as in the Caption of Fig. 1 of the original submision).
>
> As also described in the original submission version (bottom of p. 2 and Sec. 3.2), the classical feedforward network $F$ takes as input the concatenation $(z_{t-1} : x_t)$ and outputs a parameter vector $\theta_t$. Each component $\theta_t^i$ is mapped to one of the labelled parametrized gates in Fig. 1(a) (the 16 numbered $R_X/R_Y$ boxes) and sets its **rotation angle** at timestep $t$. Thus at each step we apply a single PQC $U(\theta_t)$ with a fixed gate layout but fully trainable, time-varying parameters $\theta_t$, rather than a composition $U_{\text{enc}}(x_t),U_{\text{rec}}$ with a distinct encoder $U_{\text{enc}}(x_t)$.
>
> We will point this out more explicitly in the rebuttal revision.
>
> Regarding the question of “what portion of the model’s expressivity truly arises from quantum effects,” our intention is not to attribute expressivity to a dedicated data-encoding layer. Rather, the expressive power of the recurrent quantum core arises from the learned sequence of unitaries $U(\theta_t)$ acting on an $n$-qubit state in a $2^n$-dimensional Hilbert space, together with mid-circuit expectation-value readouts that are fed into a nonlinear classical controller. This matches the measurement-feedback structure considered in the concurrent RQNN universality theory, where classical processing of measurement outcomes and adaptive re-parametrization of the full PQC play a central role.
>
> We will make this connection and our design choice more explicit to avoid any further ambiguity.

---

> > ### Author Response · Authors · 2025-11-27
> > **Re: Weakness 2**
> >
> > > Modest empirical gains: Across tasks, improvements are small (e.g., +2% accuracy in classification, BLEU 29.2 --> 31.9 for German–English translation).
> >
> > We would like to clarify that we do not intend this model to be run on classical hardware as a competitive alternative to current classical architectures, nor do we position it as a near-term solution on NISQ devices. The goal of this work is not to outperform classical models under today’s conditions, but to establish viability of a feedback-driven quantum recurrent architecture. We have added the following to the uploaded rebuttal revision:
> >
> > Lines 479-480: [...] although hardware implementations for large-scale sequence modeling would require
> > fault-tolerant devices capable of sustaining long coherent recurrences and real-time classical control.
> >
> > More specifically, our aim is to show that, deliberately under idealised, noise-free simulation, the architecture (i) trains , (ii) achieves performance that is competitive with strong classical baselines, and (iii) exhibits gradient propagation properties consistent with its unitary recurrent core, across non-trivial sequence tasks. We see this as a necessary foundational step, given the nascent state of the field and the inherent limitations and cost of classical simulation of quantum circuits, rather than as an attempt to claim present-day superiority over scaled and highly optimized classical RNNs/LSTMs.
> >
> > We would revise the introduction to be more explicit about this.
> >
> > > These gains do not convincingly justify the significant simulation cost or architectural complexity, especially when comparable improvements could be achieved by scaling classical models.
> >
> > We see questions of scaling and efficiency relative to classical models as a matter for future work on (i) more efficient quantum simulators (cuQuantum & cuTensornet from nVidia) and (ii) eventual quantum implementations on fault-tolerant hardware. In particular, an $n$-qubit recurrent core operates in a Hilbert space of dimension $2^n$; matching this with a classical recurrent model that explicitly represents a hidden state of comparable dimension (e.g., $2^{200}$ for $n = 200$) would be prohibitive in both memory and computation, while on a quantum computer, it requires $O(poly(n))$ gates and $n$ qubits.
> >
> > We take from the original submission the following regarding this:
> >
> > lines 133 - 136:
> >
> > "The PQC (Sim et al., 2019) uses only elementary one- and two-qubit
> > gates that should be supported on any hardware platform. It replaces conventional recurrence with
> > expressive unitary transformations that are physically grounded. The model performs competitively
> > in simulation, providing a hardware-aware base case and a plausible path toward future hardware
> > implementations."
> >
> > lines 484 - 485:
> > As simulation techniques improve and quantum hardware matures, this points toward practical, hardware-realistic quantum models for
> > sequential learning.

---

> > > ### Author Response · Authors · 2025-11-27
> > > **Re: Weakness 3 and 4**
> > >
> > > We thank the reviewer for further comments.
> > >
> > > > wall-time comparison
> > >
> > > We will add a training wall-time comparison to the Appendix for the rebuttal revision.
> > >
> > >
> > > > The paper reports “24 measurements for 8 qubits” but does not specify whether this is per timestep or per sample,
> > >
> > > This is per-step, on each wire we do three "measurements" obtained via Pauli expectation values, combining across all the wires we obtain 24 expectation values. For n qubits, the number of expectation values is 3n.
> > >
> > > In the uploaded rebuttal revision we have replaced all references to "measurements" with expectation-value readouts which more accurately describe the per-step "measurement" scheme. We have also updated Sec 3.2 lines 228 - 235 to more precisely describe the per-step readout scheme.
> > >
> > >
> > > > nor how parameter gradients were obtained. If the parameter-shift rule was used, the number of measurements would change, significantly impacting the total computational cost. Clarification is needed to assess efficiency and scalability.
> > >
> > > We use classical backprop with Adam (lines 246 and 295 in the original submission); hence no parameter-shift rule was used.

---

> ### Author Response · Authors · 2025-11-27
> **Re: Weakness 5 and 6**
>
> > Unclear goal between physics and simulation: It remains ambiguous whether the main objective is to demonstrate a computationally advantageous quantum model or to motivate unitary evolution as a conceptual analogue for classical nonlinearities. Without a clear experimental focus, the narrative risks oscillating between hardware-motivated and purely simulated reasoning.
>
> Please kindly refer to above comments Re: Weakness 2.
>
> > Related work gap: Although several hybrid sequence models are cited, the omission of Quantum Deep Equilibrium Models is notable, as it directly addresses the encoding overhead and representational collapse issues that also affect this architecture.
>
> Conceptually, QDEQs are orthogonal to our contribution. They introduce a training paradigm (DEQs) for PQC-based models that reduces effective depth and parameter count by solving for a fixed point, and are applied to feedforward quantum classifiers on image tasks (MNIST, FashionMNIST, CIFAR). In contrast, our work focuses on the architecture and dynamics of a feedback-driven recurrent quantum model with mid-circuit expectation-value readouts and classical nonlinear control, aligned with recent universality results for RQNNs. Our PQCs are already shallow and fixed per timestep, so depth reduction and DEQ-style fixed-point training are not the core issues we study here.
>
> We will clarify in the revision that QDEQs provide a complementary, training-level technique for PQC models, whereas our contribution is to demonstrate the empirical viability of a hybrid quantum–classical recurrent architecture with measurement-based feedback. Combining DEQ-style training with such recurrent quantum models is an interesting direction for future work, but lies outside the scope of this paper. We will also add a comparison with QDEQ on MNIST in the experimental section.

---

> > ### Author Response · Authors · 2025-11-27
> > **Re: Weakness 7**
> >
> > > While conceptually creative, the paper does not provide compelling empirical or theoretical evidence that hybrid quantum-classical RNNs outperform, generalize better, or offer unique interpretability compared to classical counterparts. The design lacks clarity regarding the role of data encoding and omits key implementation details needed to evaluate its feasibility on hardware. The work’s contribution lies more in architectural exploration than in demonstrated quantum advantage.
> >
> > We appreciate the reviewer’s summary. We express that our intent is more modest and, we believe, more appropriate to the current state of the field: to introduce and study a feedback-driven hybrid quantum–classical recurrent architecture, aligned with recent universality results for RQNNs, and to establish its empirical viability on realistic sequence tasks under idealized conditions.
> >
> > On the empirical side, we do not claim that QRNNs outperform classical baselines in a dramatic way, nor that they currently generalize better in any provable sense. Instead, we show that a measurement-feedback QRNN (i) trains with backpropagation on six realistic tasks (IMDB, MNIST, pMNIST, copying memory, word-level language modelling, machine translation), (ii) achieves performance that is competitive with RNN/LSTM/scoRNN baselines under comparable parameter and training budgets, and (iii) exhibits gradient-norm behaviour consistent with its unitary recurrent core. We have now added visualizations of the intermediate hidden states for QRNN vs. LSTM vs. scoRNN, which reveal qualitatively distinct temporal structure and provide an initial interpretability angle on how the quantum recurrent core behaves.
> >
> > On the theoretical side, we do not attempt to prove that our PQC implementation is strictly more expressive or generalizes better than all efficient classical unitary/orthogonal RNNs. Instead, incidentally our model concurrents to a universality theory (see top of page) showing that recurrent quantum networks with measurement-based feedback can approximate broad classes of state-space systems and fading-memory filters; our architecture instantiates precisely a form of this feedback structure with a coherent quantum state as recurrent memory and a nonlinear classical controller. We will make this connection more explicit in latter revisions.
> >
> > Regarding “clarity of design” and implementation details: we have clarified above that (i) we do not separate a fixed data-encoding block from a trainable PQC but the entire parametrized ansatz is controlled by the classical network at each timestep; (ii) mid-circuit “measurements” are implemented as exact expectation values of single-qubit Pauli observables, with 3 × $n$ readouts per timestep (for $n$ qubits); and (iii) all gradients are obtained by automatic differentiation via backprop and Adam, without sampling-based estimators/parameter shift rules. We also explicitly scope our claims to ideal, noise-free simulation and explain why we view this as a necessary first step before hardware feasibility or noise-robustness can be meaningfully assessed, especially in the absence of a practical quantum backpropagation method.
> >
> > In summary, we agree that the main contribution of this work lies in architectural exploration and empirical viability, rather than in demonstrating a current quantum advantage. Our results provide a concrete, learnable instantiation of a feedback-driven quantum recurrence that can serve as a foundation for future work on scaling and hardware implementations, and to experimental studies regarding "the RQNNs are able to approximate regular state-space systems without the curse of dimensionality, using quantum circuits with qubit number only growing logarithmically in the reciprocal of the prescribed approximation accuracy.", which points to exponential incarease in accuracy with only log growth in the number of qubits.

---

> > > ### Author Response · Authors · 2025-11-27
> > > **Re: Questions**
> > >
> > > We appreciate the reviewer's further questions.
> > >
> > > > In this type of architecture, reading and feeding back to the QCP could have a great cost; is it the case for this QML model too?
> > >
> > > The main cost come from simulating the PQC especially when increasing qubit count.
> > >
> > > The classical simulation toolchains are at a nascent stage, with the quantum-equivalent of nVidia CUDA being actively developed (cuQuantum) to facilitate testing ideas in simulation (again beyond moderate qubit count, a quantum computer is needed). TorchQuantum is currently the most efficient publicly available toolkit, especially for dynamic computation graphs which are important for recurrent nets. Classical RNNs/LSTMs all have highly optimized C++ CUDA kernels in Torch, which are not available for the quantum circuits. For example, on IMDB, classical LSTM takes less than ~10secs/epoch, while with QRNN, it takes ~10mins on IMDB (with 8 qubits) for example, with 12 qubits blow up VRAM on moderately sized GPUs. On the MT data eg, the max qubit count is less than 15 or so on a single 80GB GPU, with qubits close to the limit becoming prohibitively slow to train. There is currently no multi-GPU support on the public release of TorchQuantum. Another aspect is Complex Number-related ops (which are essential for classical simulation) in pytorch are WIP (e.g., see https://github.com/pytorch/pytorch/issues/125718).
> > >
> > >
> > >
> > > > Are canonical initialization schemes (e.g., amplitude or angle encoding) used for classical input embeddings?
> > >
> > > Angle encoding is used, as explained in the paper (please kindly refer to our response to Weakness 1 above.
> > >
> > > > Is the reported “24 measurements for 8 qubits” the number of observables per timestep or per sample? For training, how was gradient estimation done (e.g., via parameter-shift rule)?
> > > > How do the runtime costs per epoch compare quantitatively against the classical RNN baseline (e.g., same hardware and dataset)?
> > >
> > > Please see above Re: Weakness 3 and 4.
> > >
> > >
> > > > Will replacing the PQC with an orthogonal matrix RNN test whether unitarity alone explains observed gains?
> > >
> > > Please see RE: Weakness 1 above.

---

### Official Review · Reviewer_YrVD · 2025-10-31

**Soundness:** 2
**Presentation:** 2
**Contribution:** 2
**Rating:** 2
**Confidence:** 4

**Summary:**

In this work, the authors propose a hybrid RNN in which the recurrent core is a PQC and the classical part comprises a small feedforward network. The quantum state itself is treated as the RNN hidden state and evolved unitarily, which the authors claim naturally yields norm-preserving recurrence and, hence, better gradient propagation. To obtain per-timestep output but keep quantum memory, they simulate mid-circuit measurements/projective readouts and feed the measured values back to the classical controller for the next step. Their empirical results on six sequence tasks show that QRNN variants are competitive with RNNs, LSTMs, and scoRNNs of roughly similar parameter count.

**Strengths:**

1. The proposed quantum model is run across several/realistic sequence tasks instead of just MNIST or toy memory tasks.

2. The authors give hyperparameters, optimizer, qubit counts, measurement sets, and even report variability across 50-100 runs in the appendix.

**Weaknesses:**

1. All results are obtained in TorchQuantum on GPUs, and there is no real hardware, no noisy simulator, no demonstration that the mid-circuit readout trick they rely on can actually be executed at the depth/width they need. The paper itself admits that it models mid-circuit measurement “as a limiting case” and that present toolchains are “less optimized” for hybrid recurrence.

2. The paper leans heavily on: “the PQC is unitary ⇒ norm-preserving ⇒ better gradients ⇒ better long-sequence learning.” But: (a) We already have unitary/orthogonal RNNs (Arjovsky et al. 2016; Jing et al. 2019) that give this without simulating a quantum circuit; (b) Their own best results require adding classical nonlinearities (ReLU, GELU, GLU) in the controller, so that the actual performance bump seems to come from the classical part, not the quantum recurrence. They even show QRNNLinear is clearly worse. So the “quantum as recurrent memory” story is blurred; (c) They do not show that the PQC is doing something strictly more complex than an (efficient) unitary RNN.

3. Although the authors cite Bausch 2020, QRNN-like PQC recurrences, and QLSTM variants, the novelty is not tight compared with prior quantum-RNN / quantum-RL / hybrid quantum-classical NN.

**Questions:**

1. Can the authors show one nontrivial sequence task (longer than 400 steps, or multi-turn translation) where all classical baselines degrade but your QRNN still trains?

2. Can the authors show a side-by-side with a classical unitary/orthogonal RNN that has the same number of parameters as your PQC (counting gates) and the same measurement dimension?

3. Can the authors cost and prototype the mid-circuit measurement pipeline on any real backend (even for 4–6 qubits) to demonstrate that the feedback loop is not purely a simulator artifact?

---

> ### Author Response · Authors · 2025-11-26
> **Re: Weakness 1**
>
> We sincerely thank the reviewer for comments, and we address each of your concerns below.
>
> > All results are obtained in TorchQuantum on GPUs, and there is no real hardware, no noisy simulator, no demonstration that the mid-circuit readout trick they rely on can actually be executed at the depth/width they need.
>
> (In addition to the following comment, we kindly refer the reviewer to the top of this page for our comments regarding this weakness.)
>
> We agree that our experiments are purely classically simulated, and we are explicit about this in the paper. This is deliberate rather than an oversight. Our goal here is not to claim an immediate NISQ-era hardware advantage, but to establish that the feedback-driven quantum recurrent architecture is (i) well-posed, (ii) trainable with gradient methods, and (iii) competitive with strong classical RNN/LSTM baselines on broad-spectrum realistic sequence tasks, under noiseless simulations. In our view, this is a necessary first step before it is meaningful to discuss scaling on fault-tolerant hardware. If a model does not demonstrate viability in noiseless simulations, which provide the most favourable possible setting for learning, then adding hardware noise (at inference time only with the lack of scalable quantum backprop atm) can only degrade its behaviour further, and the experiments more convoluted, making it unclear how such an architecture could ever be made plausible on real devices.
>
> > The paper itself admits that it models mid-circuit measurement “as a limiting case” and that present toolchains are “less optimized” for hybrid recurrence.
>
> In our implementation, “mid-circuit measurement” is not a heuristic trick but the exact expectation value of Pauli observables under the simulated state. We have clarified this at all places in the uploaded revision where we replaced all references to projective measurements with "expectation values".
>
> This corresponds to the primitive assumed in the concurrent universality theory we cite: at each step, information extracted from the quantum state is fed back to control later unitaries. Our “limiting case” phrasing is to emphasize that we work in the noise-free limit and maintaining coherence of the quantum state across time; mathematically, the mid-circuit readouts and feedback are implemented exactly for the state-vector simulator. There are also two practical ways to relax this limit: 1) by the universality proof, we can do projective measurements and reinitialize at the next timestpe, and 2) as we suggested in the discussion section "via ancilla-mediated schemes in which auxiliary qubits are entangled with the main circuit, measured, and reset as
> needed while the recurrent memory remains coherent. This aligns with mid-circuit measure-and-reset
> operations already supported on several platforms (DeCross et al., 2022; Lis et al., 2023; Norcia
> et al., 2023), although hardware implementations for large-scale sequence modeling would require
> fault-tolerant devices capable of sustaining long coherent recurrences and real-time classical control."
>
> Further, introducing ad-hoc noise channels (with platform-dependent parameters) on top of our simulation would not tell us whether the architecture itself is learnable: such noise would be injected only at inference, not during training, and there is currently no practical quantum backpropagation method that would allow us to train the same model natively on hardware. As a result, noisy forward-only simulations would conflate architectural viability with arbitrary implementation details of a particular device, while still leaving the central question "can this feedback-driven QRNN be trained effectively", unanswered. We therefore, like many experimental studies, focus on the clean, ideal setting and treat noise as an orthogonal, future engineering problem once training-capable hardware is available.
>
>
> > present toolchains are “less optimized” for hybrid recurrence
>
> We have to clarify that we do not intend the model to be run on classical computers to compete with current models or even on NISQ devices. When we say that present toolchains are “less optimized” for hybrid recurrence, this is not a weakness of the model but of the current quantum-simulation ecosystem: in contrast to classical RNNs/LSTMs with highly tuned CUDA kernels, state-vector simulators for dynamic, mid-circuit-feedback PQCs are still largely catching up.
>
> In short, while this paper does not demonstrate an immediate hardware-level speedup, it does demonstrate that the specific feedback-driven quantum recurrent mechanism identified in recent theory can be instantiated, trained, and scaled to realistically sized sequence tasks under ideal conditions. We believe this “empirical proof” is precisely the kind of result that is complementary to future work that will study noise, compilation, and hardware-specific deployments once quantum backpropagation and fault-tolerant devices become available.

---

> > ### Comment · Reviewer_YrVD · 2025-11-28
> > **Follow-up with the authors' rebuttal responses**
> >
> > I thank the authors for their detailed rebuttal. While I appreciate the clarifications and the improved exposition, several of the core concerns raised in the initial review remain only partially addressed.
> >
> > 1. Although the rebuttal repeatedly emphasizes that the work does not aim at the hardware level speedup and highlights a proof-of-principle in idealized simulation, the results still do not show that the PQC recurrence confers a unique benefit that purely classical architectures cannot match. In particular, the performance bump seems to come from the classical nonlinear controller, and QRNNLinear performs notably worse, which shows the PQC alone is not competitive. Moreover, no ablation addresses the absence of a concrete scenario in which the quantum component is necessary. Thus, the main objective ("the quantum recurrence is not clearly doing something nonclassically beneficial") remains only partially addressed.
> >
> > 2. In the authors' rebuttal, they assert that expectation-value readouts are a valid limiting case. However, the paper still models non-physical operations, such as zero-noise, instantaneous feedback, and the expectation value of unlimited precision. No approximating scheme (sampling overhead, shot noise impact, latency constraints) is quantified; no small-case real backend experiment is attempted. Thus, the concern that the key mechanism lacks plausible hardware realizability remains only partially resolved.
> >
> > 3. Although the authors list differences, the distinctions remain incremental rather than fundamentally novel. The provided prior work also combines PQC recurrences with measurement-based feedback, and this work mainly substitutes expectation values for sampled measurements and adds a classical controller. Furthermore, there is no clear demonstration that the architecture achieves something qualitatively new, and it merely matches baselines on standard tasks.
> >
> > 4. IMDB statistics in Table 11 show extremely high variance (failed runs below 70% accuracy), and the quantum barren-plateau concerns may reappear at larger scales.
> >
> > In conclusion, he rebuttal clarifies, but does not convincingly resolve the central issues regarding the necessity of the quantum component, hardware feasibility, the strength of novelty, and empirical robustness. Therefore, I maintain my evaluation score and cannot raise the score at the moment.

---

> ### Author Response · Authors · 2025-11-26
> **Re: Weakness 2**
>
> We thank the reviewer again for further comments. We address them below.
>
> > The paper leans heavily on: “the PQC is unitary ⇒ norm-preserving ⇒ better gradients ⇒ better long-sequence learning.”
>
> We agree that unitary/orthogonal RNNs (e.g., Arjovsky et al. 2016; scoRNN 2018) already exploit norm preservation to improve gradient behaviour in long sequences, and we do not claim that “unitary ⇒ better gradients” is a uniquely quantum insight. But we do get 3 things for "free" in this QRNN: 1) complex-valued unitary state and state transformations; 2) exponentailly-sized state in the complex Hilbert space; and 3) norm-preservation by definition of the PQC. Moreover, we instantiate a quantum-grounded recurrent mechanism that (i) aligns with recent universality results for feedback-driven quantum RNNs, and (ii) can, in principle, leverage exponentially large Hilbert spaces on quantum hardware while maintaining unitary recurrence by construction. We also show its empirical behaviour across a broad class of sequence learning tasks.
>
>
> > But: (a) We already have unitary/orthogonal RNNs (Arjovsky et al. 2016; Jing et al. 2019) that give this without simulating a quantum circuit;
>
> We'd like to kindly clarify that the difference is
> architectural and asymptotic: the QRNN's hidden state is an $n$-qubit quantum state in a
> $2^{n}$-dimensional Hilbert space, with dynamics implemented by compact parametrized
> circuits that are natively unitary. This gives a route (with current concrete empirical experiments as a base case) to
> manipulating very high-dimensional recurrent states (e.g., effective hidden size
> $2^{200}$) with only $O(\mathrm{Poly}(n))$ quantum gates and $n$ qubits, whereas simulating an
> equivalent classical orthogonal RNN would require explicitly storing and updating
> vectors and matrices in that full space. We do not claim to demonstrate a separation at
> the small scales we simulate, but we do provide a concrete architecture that matches the
> structure assumed in the concurrent universality theory and is designed to be realisable
> on future hardware, with the PQC core being completely hardware realistic.
>
> > (b) Their own best results require adding classical nonlinearities (ReLU, GELU, GLU) in the controller, so that the actual performance bump seems to come from the classical part, not the quantum recurrence. They even show QRNNLinear is clearly worse. So the “quantum as recurrent memory” story is blurred;
>
> We fully agree that the nonlinear classical controller is crucial, and this is not a bug but a feature of the *hybrid* design. One of contributions here is to couple nonlinearity with a measurement-based feedback mechanism.  Our *QRNNLinear* ablation is not intended to show that ``the quantum part is useless''; rather, it demonstrates that a
> purely linear recurrent system, even with a quantum state, requires nonlinearity just like classical (unitray) RNNs/LSTMs do. And we show using a nonlinear classical controller benefits simplicity and efficiency compared with quantum emulations of (classical) nonlinearities within PQCs. More specifically, the full QRNN uses a coherent quantum state as recurrent memory and a nonlinear classical controller to (i) extract mid-circuit expectation-value readouts, and (ii) adaptively reparametrize the PQC at each step. This precisely matches the measurement-based feedback mechanism identified as
> structurally essential for universality in the concurrent RQNN theory, although we futrhre show where classical
> processing of measurement outcomes is also indispensable. *This has not been demonstarted before, to the best of our knowledge.*
>
> > (c) They do not show that the PQC is doing something strictly more complex than an (efficient) unitary RNN.
>
> We do not assert a proven superior complexity over classical orthogonal RNNs, but instead position our work as an empirical, architecture-level realisation of a feedback-driven QRNN model, and as a viable candidate for future quantum implementations. If anything, our architecture is simple and compact in the use of the PQC chosen, and in the feedback-based integration of a non-linear classical controller.

---

> ### Author Response · Authors · 2025-11-26
> **Re: Weakness 3**
>
> We thank the reviewer for sharing this concern.
>
> > Although the authors cite Bausch 2020, QRNN-like PQC recurrences, and QLSTM variants, the novelty is not tight compared with prior quantum-RNN / quantum-RL / hybrid quantum-classical NN.
>
> We would kindly refer to the top reply for all reviewers and reiterate the following regarding contributions and novelty.  We would appreciate if the reviewer could provide pointers to prior work where one more more of the following had been shown.
>
> - A new hybrid recurrent mechanism that combines 1) coherent quantum memory with 2) nonlinear classical control and 3) mid-circuit readouts in a novel feedback scheme.
>
> - The first empirical demonstration of such a quantum model (RNN or otherwise) across realistic sequence-learning tasks (at the scaling limits of single-gpu simulations and limitations of current toolchains). We also demonstrate for the first time measurement readouts can be successfully used in soft attention sequence2sequence learning.
>
> - Our architecture provides a concrete, trainable recurrent mechanism identified in recent concurrent theory, linking universality results to an actual implementable design.
>
> These contributions establish the first conceptual motivation and the practical viability of a hybrid QRNN, aligned with a universality theory (https://openreview.net/forum?id=248ysaRatx).

---

> ### Author Response · Authors · 2025-11-26
> **Re: Questions 1-3**
>
> > Can the authors show one nontrivial sequence task (longer than 400 steps, or multi-turn translation) where all classical baselines degrade but your QRNN still trains?
>
> We view this as an interesting direction but beyond the scope of the present work, and, importantly, not required for the claims we actually make. To our knowledge, this is the first quantum model shown empirically on all the full-scale tasks we considered here.
>
> First, our current benchmarks already include genuinely long-sequence behaviour under full
> backpropagation: on IMDB we train with untruncated backpropagation through sequences of up
> to 400 tokens for all models, and on the copying memory task each sequence contains
> $T + 20$ steps with $T = 200$. In both settings the QRNN trains and achieves
> performance competitive with RNN/LSTM/scoRNN baselines. We also evaluate on full-scale
> MNIST/pMNIST, where the recurrent core must integrate information across the full 2D digit,
> and on sequence-to-sequence machine translation with an attention mechanism over
> mid-circuit readouts. These tasks are deliberately chosen to combine realistic structure
> (natural language, images, seq2seq) with the simulation constraints of current quantum
> toolchains.
>
> > Can the authors show a side-by-side with a classical unitary/orthogonal RNN that has the same number of parameters as your PQC (counting gates) and the same measurement dimension?
>
> In all the experiments, it's the case that the baseline hidden dimensions are strictly greater than the number of measurements as a result of matching the parameter count of the baselines and QRNN (as in the experimental tables). We have updated the scoRNN results in all the experiments with scoRNN parameter count matching the QRNN.
>
> > Can the authors cost and prototype the mid-circuit measurement pipeline on any real backend (even for 4–6 qubits) to demonstrate that the feedback loop is not purely a simulator artifact?
>
> Please kindly refer to our comments above Re: Weakness 1.
>
> We thank the reviewer again for all the comments and questions.

---

> ### Author Response · Authors · 2025-11-28
>
> > Although the rebuttal repeatedly emphasizes that the work does not aim at the hardware level speedup and highlights a proof-of-principle in idealized simulation, the results still do not show that the PQC recurrence confers a unique benefit that purely classical architectures cannot match. In particular, the performance bump seems to come from the classical nonlinear controller, and QRNNLinear performs notably worse, which shows the PQC alone is not competitive. Moreover, no ablation addresses the absence of a concrete scenario in which the quantum component is necessary. Thus, the main objective ("the quantum recurrence is not clearly doing something nonclassically beneficial") remains only partially addressed.
>
> Thanks again to the reviewer for engaging in further discussion and comments.
>
> We'd like to clarify that the scope of this work is to introduce a feedback-driven QRNN, and to show that it is trainable and competitive with strong classical baselines under ideal noiseless simulation.
>
> - The PQC recurrence is quantum-native: it is built entirely from one- and two-qubit gates, and it is can run directly on quantum hardware.
>
> - The PQC Hilbert space grows exponentially (2^n) with O(poly(n)) growth of gates, with n qubits only.
>
> - The removal of nonlinearities was an ablation experiment. Likewise, if one removes the nonlinearities from an LSTM, one would expect a performance drop; similarly for removing the nonlinearities from the feedforward blocks in a Transformer. However, in those cases we do not conclude that the other parts of the models are “useless”.
>
> - It was not within the scope of this paper, nor part of our claim, that the PQC alone would be competitive. We propose a hybrid architecture in which the classical nonlinear controller facilitates a feedback mechanism and naturally lives on a classical device, even when the recurrent core is implemented on quantum hardware.
>
> > In the authors' rebuttal, they assert that expectation-value readouts are a valid limiting case. However, the paper still models non-physical operations, such as zero-noise, instantaneous feedback, and the expectation value of unlimited precision. No approximating scheme (sampling overhead, shot noise impact, latency constraints) is quantified; no small-case real backend experiment is attempted. Thus, the concern that the key mechanism lacks plausible hardware realizability remains only partially resolved.
>
> We would kindly refer to Re: Weakness 1 above. We view noiseless simulation as an essential first step, abstracting away from platform-dependent details, providing a clean baseline that can inform realistic hardware implementations.
>
> > Although the authors list differences, the distinctions remain incremental rather than fundamentally novel. The provided prior work also combines PQC recurrences with measurement-based feedback, and this work mainly substitutes expectation values for sampled measurements and adds a classical controller.
>
> We thank the reviewer for this helpful comment and would be very grateful if they could kindly point us to the specific prior work, as we are not currently aware of existing architectures that combine PQC recurrences with measurement-based feedback in the way we do here.
>
> > Furthermore, there is no clear demonstration that the architecture achieves something qualitatively new, and it merely matches baselines on standard tasks.
>
> We respectfully note that, beyond matching baselines on standard tasks, our results also indicate qualitatively new behaviour: we show quantitatively improved gradient propagation for sequences up to 400 tokens (Fig. 3), and qualitatively distinct temporal structure in the hidden-state visualizations added in Appendix B of the rebuttal revision. None of which had been shown before in a quantum rnn.
>
> > IMDB statistics in Table 11 show extremely high variance (failed runs below 70% accuracy), and the quantum barren-plateau concerns may reappear at larger scales.
>
> We kindly note this was not mentioned in the original review. But we agree that Table 11 shows noticeable variance, and we reported many seeds precisely to be transparent about this. But this doesn't diminish the claims of the work. Importantly, Fig. 3 shows that gradient norms remain well above numerical noise even for sequences of length 400, so we do not see the exponential gradient collapse typically associated with barren plateaus at the depths and qubit counts considered here. The hyperparameters (especially Adam-related) were tuned primarily on the classical baselines for efficiency; the QRNNs reuse these settings with only light adjustment, which leaves room for QRNN-specific tuning, with better simulation toolchains.

---

### Official Review · Reviewer_VTBd · 2025-11-03

**Soundness:** 3
**Presentation:** 4
**Contribution:** 3
**Rating:** 4
**Confidence:** 3

**Summary:**

This paper presents a novel Hybrid Quantum-Classical Recurrent Neural Network (QRNN). The work is one of the first demonstrations of a quantum-grounded model achieving competitive or superior performance against classical baselines across a broad and realistic (but toy) suite of sequence-learning tasks. The paper is well-written, but I do not have enough expertise to properly evaluate this paper and I made an educational guess.

**Strengths:**

- The model is evaluated on six diverse tasks (sentiment analysis, MNIST, pMNIST, copying memory, language modeling, machine translation) and is shown to be competitive with or outperform classical RNNs, LSTMs, and specifically designed orthogonal RNNs (scoRNN).

- The paper is generally well-structured and clearly written. I enjoyed the reading flow.

- The paper thoughtfully discusses the path to hardware implementation, acknowledging current simulation limits and proposing a realistic ancilla-mediated measurement scheme for future work. The choice of a simple, hardware-native PQC ansatz strengthens the practical relevance of the work.

**Weaknesses:**

- What is the advantage  of Hybrid Quantum-Classical Recurrent Neural Network? comparing to Classical RNN or  other Hybrid quantum NN (Like Hybrid quantum CNN if one can implement)?
- It would be nice if the authors could explain something related to GPU consumption or efficency.
- It would be nice if the authors could visualize something related to the intermediate hidden states, like the state change in QRNN.
- To which perspective does the design of QCNN could  benefit the majority of  ICLR audiences.
- The reviewer would appreciate it if the authors could provide some discussions on existing work like Li et.al. https://arxiv.org/pdf/2302.13812 and many other quantum-classifical hybrid models.

**Questions:**

see the weakness.

---

> ### Author Response · Authors · 2025-11-26
> **Re: Q1**
>
> We sincerely appreciate the reviewer's questions and the opportunity to clarify.
>
>
> > What is the advantage of Hybrid Quantum-Classical Recurrent Neural Network? comparing to Classical RNN or other Hybrid quantum NN (Like Hybrid quantum CNN if one can implement)?
>
>
> We would like to emphasize that we do not view this architecture as a (quantum-inspired) classical model. It is a genuinely hybrid design whose quantum recurrent core is intended to be realised on quantum hardware. Many standard quantum algorithms, such as Shor’s factoring algorithm, combine quantum subroutines with classical computation, where some steps are run on a quantum device and others on a classical processor, and the two interact to progress the computation. Having a classical component in the architecture does not diminish its quantum nature; it reflects the practical reality of how quantum systems interface with classical data and control mechanisms.
>
> In our QRNN, the recurrent core is a parametrized quantum circuit whose hidden state is a quantum state living in a Hilbert space of dimension $2^n$, where $n$ is the number of qubits. This space grows exponentially with $n$, and beyond moderate sizes (e.g., $n = 200$), it becomes classically intractable to store or manipulate. A quantum computer (with superposition and entanglement, both of which our PQC gates in Fig. 1 of the paper naturally utilise, where each qubit is in a superposition, and the two qubit gates entangle them), however, can efficiently operate in this space via superposition, entanglement, and compact $n$-qubit parametrized circuits. The PQC is also unitary by construction, and a potentially exponentially large representational space and unitary recurrence are attractive for recurrent sequence modelling, as they relate to well-known issues in sequence learning.
>
> Our QRNN combines these ingredients with mid-circuit feedback: at each timestep, mid-circuit expectation-value readouts are passed into a nonlinear classical controller, which in turn adaptively reparametrizes the PQC. This yields a coherent hybrid recurrent architecture that embeds quantum memory with classical nonlinearity. The PQC design we use is basic, and becomes classically *unsimulatable* once the qubit count is sufficiently large, providing a potential path for advantage over classical RNNs in addition to unitary dynamics.
>
> As noted in our general response to reviewers, a recent concurrent work in this conference provides a mathematical proof that measurement-based feedback is essential for universality in quantum recurrent models. To our knowledge, no comparable universality result exists for quantum CNNs (although such a proof does exist for quantum feedforward networks [1]). We are also not aware of quantum CNNs (not quantum inspired) being applied to sequence modelling across multiple realistic tasks, especially sequence-to-sequence settings such as machine translation. In this regard, we should the readouts could be used successfully in a soft attention scheme, and to our knowledge this had not been shown before in any quantum models with a path for implementation on quantum hardware supported by a universality theory.
>
>
> [1] https://arxiv.org/abs/2307.12904

---

> ### Author Response · Authors · 2025-11-26
> **Re: Q2**
>
> >It would be nice if the authors could explain something related to GPU consumption or efficency.
>
> The classical simulation toolchains are at a nascent stage, with the quantum-equivalent of nVidia CUDA being actively developed (cuQuantum) to facilitate testing ideas in simulation (again beyond moderate qubit count, a quantum computer is needed). TorchQuantum is currently the most efficient publicly available toolkit, especially for dynamic computation graphs which are important for recurrent nets. Classical RNNs/LSTMs all have highly optimized C++ CUDA kernels in Torch, which are not available for the quantum circuits. For example, on IMDB, classical LSTM takes less than ~10secs/epoch, while with QRNN, it takes ~10mins on IMDB (with 8 qubits) for example, with 12 qubits blow up VRAM on moderately sized GPUs. On the MT data eg, the max qubit count is less than 15 or so on a single 80GB GPU, with qubits close to the limit becoming prohibitively slow to train. There is currently no multi-GPU support on the public release of TorchQuantum. Another aspect is Complex Number-related ops (which are essential for classical simulation) in pytorch are WIP (e.g., see https://github.com/pytorch/pytorch/issues/125718).

---

> ### Author Response · Authors · 2025-11-26
> **Re: Q3**
>
> > It would be nice if the authors could visualize something related to the intermediate hidden states, like the state change in QRNN.
>
> We appreciate the reviewer for this suggestion.
>
> We have now added visualizations in the uploaded revised version for the rebuttal period. We also added a short description at end of Sec 4.6 and the following to the new Appendix B:
>
> For comparability across models with different dynamic ranges, each unit’s trajectory is $z$-scored
> over time (per unit) and displayed using the same color scale (clipped to $[−3, 3]$). Qualitatively, the
> QRNN exhibits more coherent, banded temporal structure (units with sustained positive/negative
> excursions across multiple timesteps), whereas the LSTM and scoRNN activations appear more
> spatially fragmented and locally varying.

---

> ### Author Response · Authors · 2025-11-26
> **Re: Q4**
>
> > To which perspective does the design of QCNN could benefit the majority of ICLR audiences.
>
> One perspective on quantum computing relevant to our work is that it can be viewed as linear algebra with complex numbers in unitary matrices/tensors, where a quantum computer naturally excels at manipulating such tensors so large that they would exceed any classical computational capacity. Complex numbers (which arise naturally in parametrized quantum circuits) and large-capacity complex tensors are both potentially very closely related to representation learning.
>
> Classical RNNs compress sequences into fixed-size states in $\mathbb{R}^h$
> with capacity bounded linearly by parameters, while Transformers address this bottleneck through attention mechanisms, though a recent line of work ([1-4] etc.) suggests that explicit recurrent structure can be beneficial for certain tasks. Our architecture investigates a complementary direction by maintaining a quantum state in an exponentially large Hilbert space $\mathbb{C}^{2^n}$.
>
> We are encouraged that our work provides a concrete, trainable implementation of the feedback-driven quantum recurrent architecture identified in recent concurrent theoretical work as necessary for universality. That work demonstrates that "feedback from intermediate measurements" is a "structurally essential ingredient" for universal approximation in quantum recurrent models. By realizing this principle through mid-circuit expectation-value readouts combined with classical nonlinear feedback control, we hope our work may contribute to the community's ongoing exploration of diverse computational primitives for sequential learning in addition to classical recurrent nets and transformers.
>
> With fault tolerant quantum computing promised by the end of this decade [0], we hope it is also timely to explore alternative approaches to recurrent learning and sequential learning in general, and our work demonstrates the viability of a quantum-grounded recurrent model. Realising the full potential of such architectures will require more mature simulation toolchains (to test ideas classically, see reply to GPU question above), or future techniques such as quantum backpropagation and fault-tolerant quantum hardware if one wishes to go beyond the moderate qubit counts we use here.
>
>
> [0] https://www.quantinuum.com/blog/technical-perspective-by-the-end-of-the-decade-we-will-deliver-universal-fault-tolerant-quantum-computing
>
> [1] https://arxiv.org/abs/2312.00752
>
> [2] https://proceedings.mlr.press/v202/orvieto23a.html
>
> [3] https://openreview.net/forum?id=6HUJoD3wTj
>
> [4] https://proceedings.neurips.cc/paper_files/paper/2024/file/c2ce2f2701c10a2b2f2ea0bfa43cfaa3-Paper-Conference.pdf

---

> ### Author Response · Authors · 2025-11-26
> **Re: Q4 II**
>
> > To which perspective does the design of QCNN could benefit the majority of ICLR audiences.
>
> We thank the reviewer again for this question. We have now added another paragraph (in the uploaded revision) to bridge the QRNN to one more perspective relevant to the ICLR audience. We copy this added paragraph here:
>
> Another way to view the QRNN is via fast and slow weights in RNNs, which function as different types
> of memory across multiple timescales (Schmidhuber, 1992; Ba et al., 2016). The PQC parameters
> serve as the short-term memory, analogous to the hidden activities of classical RNNs, and are
> controlled and reconfigured at each timestep by a classical feedforward network whose slow weights
> encode the long-term memory. The quantum state, updated via unitary transformations, evolves on a
> faster timescale than the slow weights, persists across timesteps, and acts as a third, higher-capacity
> memory in the Hilbert space, retaining information that influences subsequent computation (Hinton
> and Plaut, 1987; Schmidhuber, 1993).

---

> ### Author Response · Authors · 2025-11-26
> **Re: Q5**
>
> > ... some discussions on existing work like Li et.al. https://arxiv.org/pdf/2302.13812 and other ...
>
> There are mainly two categories of "quantum" models for classical tasks: (1) quantum-inspired models (such as Li et al. referred to in the question), which are not intended to run on quantum hardware and remain classical despite having quantum-inspired components or terminology; and (2) quantum models with hybrid components for handling classical data, where the quantum parts are intended to be actual quantum operations (with the potential to benefit from actual quantum hardware, not just quantum-inspired).
>
> Specifically, the model of Li et al. is a complex-valued classical Transformer, which is strictly a classical model that replaces real numbers with complex numbers (type 1 above). In contrast, there are also quantum "Transformers" based on purely quantum primitives/operations that have been demonstrated to run (at inference) on quantum hardware (type 2 above), e.g., [0].
>
> For RNNs, for example [1, 2 etc.] and our work are all type 2 (and Li et al. (2023) and Siemaszko et al. (2023) discussed in the related work section among others), while [3,4 etc.] are type 1.
>
> [0] Quantum vision transformer (https://quantum-journal.org/papers/q-2024-02-22-1265/pdf/)
>
> [1] Bausch 2020, Recurrent quantum neural networks
> (https://proceedings.neurips.cc/paper/2020/hash/0ec96be397dd6d3cf2fecb4a2d627c1c-Abstract.html)
>
> [2] Nikoloska et al., Time-warping invariant quantum recurrent neural networks via quantum-classical adaptive gating
>
> [3] Chen et al. Quantum Long Short-Term Memory (https://arxiv.org/abs/2009.01783)
>
> [4] Ubale et al. Toward practical quantum machine learning: A novel hybrid quantum lstm for fraud detection  (https:
> //arxiv.org/abs/2505.00137)
>
> For the rebuttal revision given space limitation, we will expand the related work section in later revisions to further clarify these distinctions.

---

### Author Response · Authors · 2025-11-20
**Addressing some common concerns**

We thank the reviewers for their feedback, and we first address some common concerns.

**Measurement- and Feedback-driven nonlinear classical control and Universality**

Concurrent to our work, a theoretical paper submitted under the same conference (https://openreview.net/forum?id=248ysaRatx) on feedback-driven quantum recurrent network shows that recurrent quantum neural networks equipped with measurement-based feedback form a universal function approximator, showing that: “1) **RQNNs are able to approximate regular state-space systems without the curse of dimensionality, using quantum circuits with qubit number only growing logarithmically in the reciprocal of the prescribed approximation accuracy**. 2) ... we prove that RQNNs can uniformly approximate the arbitrary fading memory, causal, and time-invariant filters. In particular, RQNNs have approximation properties as competitive as those of popular reservoir computing/statespace system families like echo state networks, state-affine systems, or linear systems with polynomial/neural network readouts. *These analyses identify feedback from intermediate measurements as a structurally essential ingredient: universality requires that information extracted from the quantum system be re-injected into the recurrent update*.”

Our hybrid QRNN is the first proposal (to our knowledge) realizing an architecture precisely akin to this principle. Instead of measure and reinitialise as in the theory, we maintain coherence by extracting expectation-value readouts which are used in the recurrent feedback. Our work also provides the first empirical demonstration that a feedback-driven quantum recurrent model can be trained successfully on realistic sequence-learning tasks. Prior QNN or QRNN works have been restricted to narrow or downsampled benchmarks, whereas we evaluate the hybrid QRNN on six full-scale tasks including IMDB, MNIST, pMNIST, copying memory, language modeling, and MT. *The goal is not to outperform classical models but to establish viability*: (deliberately) under idealized, noise-free simulation, the architecture can train stably, achieves competitive performance with strong classical baselines, and exhibits favourable gradient propagation properties consistent with its unitary recurrent core, on non-trivial tasks considering the nascent state of field including the inherent limitations of classical simulation costs.


**Empirical viability and the notion of “quantum advantage”**

We also argue as part of the motivation of the paper that classically it would be prohibitive to realise a hidden state of size, say $2^{200}$. QRNN aside, one advantage of a quantum computer is its ability to manipulate a Hilbert space in the Complex domain of size $2^n$, with only $n$ qubits (lines 86-88). Demonstrating viability in noiseless simulations with a moderate qubit count empirically would be a natural first step before scaling up to more qubits, with quantum implementations, on fault-tolerant quantum hardware. Another advantage of the quantum model is that it naturally gives rise to the complex Hilbert space while ensuring unitarity by construction.

Finally, our focus on idealized classical simulation is deliberate. Realistic hardware modelling requires specifying many device-dependent factors, none of which have a universal form across platforms. Introducing arbitrary noise models would not meaningfully inform the learnability of the architecture, especially when we await a working quantum backpropagation method and when noise can only be injected at inference rather than during scalable, realistic quantum backprop training, because such training requires quantum backprop [1]. For this reason, any analysis of hardware noise that does not affect the training procedure itself is inherently limited: it captures neither optimization behaviour nor model capacity, and therefore cannot meaningfully speak to the viability of the architecture. Our aim in this work is thus to establish a clean baseline for learnability under ideal conditions, which can then guide future investigation once training-capable fault-tolerant hardware becomes available, beyond the NISQ regime.


**Contributions and Novelty**

A new hybrid recurrent mechanism that combines 1) coherent quantum memory with 2) nonlinear classical control and 3) mid-circuit readouts in a novel feedback scheme.

- The first empirical demonstration of such a quantum model (RNN or otherwise) across realistic sequence-learning tasks and the first time a measurement readouts-based soft attention in seq2seq learning.

- Our architecture provides a concrete, trainable recurrent mechanism identified in recent concurrent theory, linking universality results to an actual implementable design.

- These contributions establish the first conceptual motivation and the practical viability of a hybrid QRNN, aligned with a universality theory (https://openreview.net/forum?id=248ysaRatx).

[1] https://arxiv.org/abs/2305.13362

---

### Author Response · Authors · 2025-12-02
**Summary and Updated Manuscript**

We sincerely appreciate the Reviewers’ thoughtful feedback. Their comments affirm the value of our contribution at the intersection of quantum computing and sequence learning. During the rebuttal, we have thoroughly responded to all reviewers' comments and questions. Below, we summarize the key strengths highlighted in the reviews:

* Novel and conceptually interesting architecture (Reviewers vQW1, VTBd, FAnR). Integrate quantum computing into machine learning is an important task (Reviewer FAnR).

* Broad and realistic empirical evaluation (Reviewers VTBd, YrVD, vQW1, FAnR).

* Thorough and transparent experimentation (Reviewers YrVD, vQW1, FAnR).

* Practical, hardware-aware perspective (Reviewers VTBd, vQW1).

* Clear, well-written presentation (Reviewers VTBd, FAnR).


We have also made revisions to the manuscript in response to reviewers' comments, as indicated below:

1. Plots for hidden state visualizations are now added in Appendix B (Reviewer VTBd). Accompanying descriptions have also been added in Sec. 4.6 and in Appendix B.
2. We have made it explicit in the Introduction that the recurrent core PQC is feasible on quantum hardware (as demonstrated in a recent work)  and clarified the overall architecture is intended for fault-tolerant hardware with efficient real-time classical control, to distinguish with hybrid models intended only on classical hardware (All reviewers).
3. Classical scoRNN baseline results (with matching parameter count) are now added to all applicable result tables (Reviewer YrVD).
4. Made the encoding scheme even more explicit in both Fig. 1 caption (Reviewer vQW1).
5. Made the usage of a single unified encoding and transformation unitary even more explicit in Fig. 1 caption and accompanying footnote (Reviewer vQW1).
6. Wall-time comparison with classical baselines now added to Appendix G (Reviewer vQW1).
7. Replaced all references throughout to "measurements" with expectation-value readouts which more precisely describe the per-step "measurement" scheme. We have also updated Sec 3.2 lines 228 - 235 to more precisely describe the per-step readout scheme (Reviewer vQW1).
8. Added a description of the QDEQ orthogonal work in relation to the ansatz design as a third paragraph in Appendix E; (note that it has MNIST best 73.68 accuracy vs. 98.03 of ours);  (Reviewer vQW1).
9. All baseline hyperparameter and experimental settings etc. have now been added to Appendix C (Reviewer FAnR).
10. Parameter counts are checked and updated across all the tables, experiments rerun and results updated when needed (Reviewer FAnR).
11. Details of the measurement operator now updated toward the end of Sec. 3.2 (Reviewer FAnR).
12. Clarified code would be released in the comment replying to the reviewer, along with experimental scripts (Reviewer FAnR).
13. Table added for the copying task in Appendix F (Reviewer FAnR).
14. Finally, we have made revisions to the writing throughout the manuscript.
15. scoRNN reimplemented in PyTorch and added results for it on the MT task.

---

### Meta-Review · Area_Chair_ZpC1 · 2026-01-07

**Summary:**

Three of four reviewers were unconvinced on the positive side. The goals of the experiments in this work need to be justified, and the implementation should be more detailed. They agreed that this work requires additional effort to meet the acceptance bar of ICLR. Thus, I am inclined not to accept this draft at this stage. Thank you for your effort and the rebuttal. It is an interesting work. I hope the input from the reviewers will help you further improve this work.

**Reviewer Concerns:**

This work has limited novelty and experimental results.

**Reviewer Scores:**

The reviewers' scores reflect the limitations in this work.

---

### Decision · Program_Chairs · 2026-01-26

Reject